# Association between cumulative exposure periods of flupentixol or any antipsychotics and risk of lung cancer

Yi Chai [1,2,9], Rachel Yui Ki Chu[1,9], Yuqi Hu[1,9], Ivan Chun Hang Lam [1], Franco Wing Tak Cheng [1], Hao Luo[2,3,4], Martin Chi Sang Wong[5], Sandra Sau Man Chan[6], Esther Wai Yin Chan [1,7], Ian Chi Kei Wong [1,7,10 ✉] & Francisco Tsz Tsun Lai [1,7,8,10 ✉]

## Abstract

**Background** Preclinical evidence suggests that certain antipsychotic medications may inhibit the development of lung cancer. This study aims to investigate the association between incident lung cancer and different cumulative exposure periods of flupentixol or any antipsychotics.

**Methods** Using electronic health records from the Hospital Authority in Hong Kong, this nested case-control study included case participants aged 18 years or older with newly diagnosed lung cancer after initiating antipsychotics between January 1, 2003, and August 31, 2022. Each case was matched to up to ten controls of the same sex and age, who were also antipsychotic users. Multivariable conditional logistic regression models were conducted to quantify the association between lung cancer and different cumulative exposure times of flupentixol (0–365 days [ref]; 366–1825 days; 1826+ days) and any antipsychotics (1–365 days [ref]; 366–1825 days; 1826+ days), separately.

**Results** Here we show that among 6435 cases and 64,348 matched controls, 64.06% are males, and 52.98% are aged 65–84 years. Compared to patients with less than 365 days of exposure, those with 366–1825 days of exposure to flupentixol (OR = 0.65 [95% CI, 0.47–0.91]) and any antipsychotics (0.42 [0.38–0.45]) have a lower risk of lung cancer. A decreased risk is observed in patients who have 1826+ days of cumulative use of any antipsychotics (0.54 [0.47–0.60]).

**Conclusions** A reduced risk of lung cancer is observed in patients with more than one year of exposure to flupentixol or any antipsychotics. Further research on the association between lung cancer and other antipsychotic agents is warranted.

## Plain language summary

Antipsychotic drugs are mainly used to treat mental illnesses. Certain antipsychotic medications, such as flupentixol, may help protect patients against lung cancer. Here, we investigated whether prolonged use of flupentixol or other antipsychotics could reduce the occurrence of lung cancer among antipsychotic users. We demonstrated that a smaller proportion of patients with one to five years and more than five years of exposure to any antipsychotics develop lung cancer compared to those with less than one year of exposure. Specifically, for flupentixol, we observed a smaller proportion of patients with one to five years of exposure develop lung cancer compared to those with less than one year. To substantiate our current findings, further studies examining other populations and specific antipsychotic agents are necessary for developing effective lung cancer prevention strategies among this high-risk population.

[1] Centre for Safe Medication Practice and Research, Department of Pharmacology and Pharmacy, Li Ka Shing Faculty of Medicine, The University of Hong Kong, Hong Kong SAR, China. [2] The Hong Kong Jockey Club Center for Suicide Research and Prevention, The University of Hong Kong, Hong Kong SAR, China. [3] Department of Social Work and Social Administration, Faculty of Social Sciences, The University of Hong Kong, Hong Kong SAR, China. [4] Sau Po Centre on Ageing, The University of Hong Kong, Hong Kong SAR, China. [5] Centre for Health Education and Health Promotion, The Jockey Club School of Public Health and Primary Care, Faculty of Medicine, The Chinese University of Hong Kong, Hong Kong SAR, China. [6] Department of Psychiatry, Faculty of Medicine, The Chinese University of Hong Kong, Hong Kong SAR, China. [7] Laboratory of Data Discovery for Health (D24H), Hong Kong Science and Technology Park, Hong Kong SAR, China. [8] Department of Family Medicine and Primary Care, School of Clinical Medicine, Li Ka Shing Faculty of Medicine, The University of Hong Kong, Hong Kong SAR, China. [9] These authors contributed equally: Yi Chai, Rachel Yui Ki Chu, Yuqi Hu. [10] These authors jointly supervised this work: Ian Chi Kei Wong, Francisco Tsz Tsun Lai. ✉email: wongick@hku.hk; fttlai@hku.hk

ung cancer remains one of the leading causes of death worldwide despite decades of local and global collaborative efforts to restrict tobacco use[1]. Antipsychotic users, who typically live with severe mental illnesses, have a notably higher crude incidence of lung cancer than the general population[2]. This elevated risk can be attributed to lower socioeconomic status, poorer health awareness and self-care, and most importantly, a higher smoking rate[3–5].

Interestingly, there is analytic evidence showing that people living with severe mental illnesses or using antipsychotics might have a lower risk of some cancers[6–8]. Apart from a potential direct protective effect from the medications, this may be attributed to several reasons. First, some genetic factors involved in the etiology of schizophrenia may act as protective agents against cancer[9]. For example, the tumor suppressor gene p53 can reduce the risk of cancer through apoptosis[9]. Second, long-term antipsychotic use is associated with negative physical consequences, leading to a shorter life expectancy in comparison to the general population[10]. This could contribute to lower cancer incidence. Furthermore, the stigma surrounding mental illnesses and antipsychotic prescriptions, coupled with low health literacy and the deprioritization of mental health care, have created obstacles for individuals seeking medical help[11]. Consequently, the underdiagnosis of diseases, including cancer, might result in an artificially lowered cancer incidence among those receiving antipsychotic treatments.

The use of second-generation antipsychotics has been increasingly common in recent years[12,13]. However, flupentixol, a first-generation antipsychotic agent, has maintained a stable presence in the clinical treatment of schizophrenia due to its efficacy against negative and affective symptoms and a lower reported rate of severe adverse drug reactions[14,15]. In terms of pharmacology, flupentixol demonstrates promising effects in reducing the risk of lung cancer[16]. In non-small-cell lung cancer, flupentixol interferes with the PI3K/AKT pathway, which is often hyperactivated in living cancer cells, angiogenesis, invasion, proliferation, and metastasis[17]. Specifically, the docking of flupentixol to the PI3Kα protein inhibits the PI3K/AKT pathway and the survival of lung cancer cells both in vitro and in vivo[16]. This reduces the phosphorylation of AKT protein in a concentration-dependent manner[16], blocking the activation of AKT protein to regulate downstream components[18].

Despite the speculation of this molecular mechanism, there has been little clinical research conducted to determine the association between flupentixol and lung cancer. There is only one case-control study showing reduced odds of lung cancer associated with antipsychotic use compared with non-use, which excluded flupentixol[19]. In addition, the authors of that study acknowledged the small number of antipsychotic users recruited, which could reduce the statistical power for ascertainment of association[19]. Hence, this current study aims to examine the association between flupentixol or any antipsychotic use and lung cancer among the population of antipsychotic users with varying degrees of exposure time. Given the aforementioned preclinical evidence and the unique mechanism of flupentixol, we hypothesize that there is an inverse association. Our study shows that patients exposed to any antipsychotics for 366–1825 days or 1826 + days have a decreased risk of lung cancer when compared to those exposed for less than 365 days. In particular, we note a reduced lung cancer risk in patients with 366–1825 days of flupentixol exposure compared to those with less than 365 days of exposure.

## Methods

**Study design and data source**. We conducted a nested case-control study using de-identified electronic medical records from the Hong Kong Clinical Data Analysis and Reporting System (CDARS). CDARS is a territory-wide database developed by the Hospital Authority (HA), a statutory body managing all public hospitals and providing healthcare services for more than 7 million Hong Kong residents[20]. CDARS has collected information on demographics, clinical diagnoses, procedures, admission and discharge records, laboratory tests, and prescriptions from inpatient, outpatient, and accident & emergency departments since 1993[21]. The quality and reliability of data from CDARS have been extensively validated by various pharmacoepidemiologic studies including studies on antipsychotic use as well as cancer[22–25]. For this study, we retrieved data from CDARS from September 01, 2022, to October 31, 2022. The data were reported using the Strengthening the Reporting of Observational Studies in Epidemiology (STROBE) reporting guideline[26].

**Ethics approval**. This study was approved by the Institutional Review Board of the University of Hong Kong/ Hospital Authority Hong Kong West Cluster (HKU/HA HKW IRB, reference number: UW 20-113). As our data were all anonymized without any personal identification information, no informed consent was required for the study.

**Selection of cases and controls**. We included patients who received prescriptions for any antipsychotics between January 1, 2001, and August 31, 2022. Specific antipsychotic agents were identified by the British National Formulary (BNF) chapter 4.2.1 (Antipsychotic drugs) and 4.2.2 (Antipsychotic depot injections). Patients who (1) had missing values on age, sex, and date of death; (2) had incorrect records (i.e., the death date was before the date of first prescription of antipsychotics); (3) had a diagnosis of lung cancer before or at the date of the first prescription of antipsychotics; or (4) had a diagnosis of lung cancer before January 01, 2003, were excluded. The years 2001 and 2002 were employed as the screening period to exclude non-incident lung cancer cases. Given the clinical information before 2001 was not available, the first prescription of antipsychotics between 2001 and 2022 was designated as the first prescription of antipsychotics during the study period for each patient.

The outcome of this study was the first diagnosis of lung cancer between January 01, 2003, and August 31, 2022. Lung cancer was identified by the ICD-9-CM codes 162.0–162.9.

Cases were patients aged 18 years or older with newly diagnosed lung cancer between January 01, 2003, and August 31, 2022. The index date for cases was the date of the first diagnosis of lung cancer.

Up to ten controls who were antipsychotic users and without a diagnosis of lung cancer before or at the index date were randomly matched for each case by sex and age in years using the incidence density sampling approach[27]. The index date for controls was designated as the date of the first diagnosis of lung cancer for their matched cases. Each patient can contribute as a control for up to four different cases[28], and thus, they could have up to four different assigned index dates. For each matching process, control candidates were excluded from the pool of controls if they did not have any prescription of antipsychotics before their assigned index date. Figure 1 shows the flowchart of cases and controls selection.

**Exposure**. All patients included in this study had at least one day of exposure to any antipsychotics before the index date. A patient might have multiple records of antipsychotic prescriptions. All antipsychotic prescriptions between the first antipsychotic use and the index date were extracted for each patient to calculate the cumulative drug exposure. The exposure period of a single record

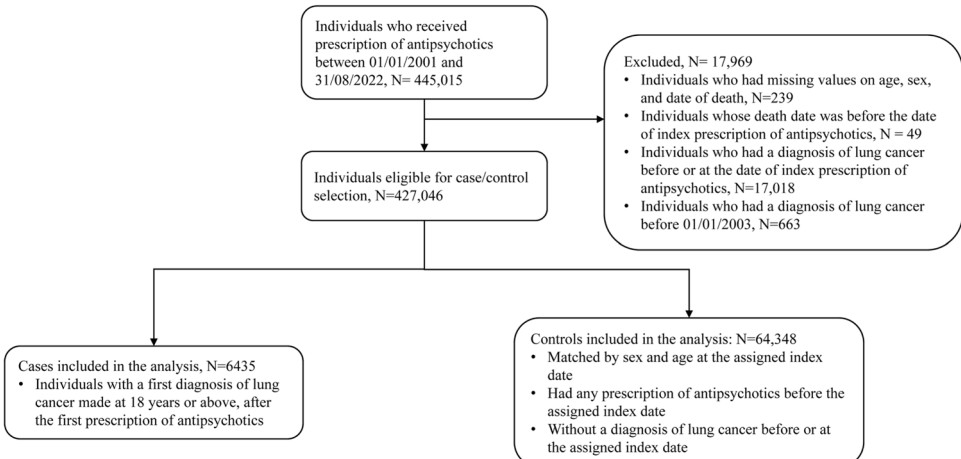

**Fig. 1 Flowchart of the study sample selection.** The index date for cases: the date of the first diagnosis of lung cancer. The assigned index date for controls: the date of the first diagnosis of lung cancer for their matched cases.

of the prescription was defined as the time between the start and the end dates of the prescription for that record. The cumulative exposure of all antipsychotics was calculated by summing up the periods of all records of any antipsychotic prescriptions before the index date. If multiple drugs were prescribed within the same period, the exposure period of all antipsychotics would be counted only once. The treatment period of flupentixol was calculated by summing up the periods of all records of flupentixol prescriptions before the index date. According to current clinical guidelines, which suggest the treatment duration of antipsychotics should be one to 5 years[29], the cumulative exposure to any antipsychotics (including flupentixol) was categorized as 1–365 days (selected as the reference group), 366–1825 days, and 1826+ days. Since not all included antipsychotic users received flupentixol, the cumulative exposure of flupentixol was categorized as 0–365 days (selected as the reference group), 366–1825 days, and 1826+ days. Two analyses were conducted separately to investigate the association between the different cumulative exposure times of (1) flupentixol; and (2) any antipsychotics and the reduced risk of incident lung cancer.

More than 98% of records had the start and end dates of antipsychotic prescriptions. We used the dosage, and prescribed quantity and frequency to determine the prescription duration when the start or end dates were not available. We further used the median prescription value of each specific drug to impute the corresponding drug exposure period when the above information was missing.

**Confounders**. We adjusted the potential risk factors of lung cancer or indications for antipsychotic use as confounders to identify the independent association between antipsychotics and lung cancer, including the use of non-steroidal anti-inflammatory drugs (NSAIDs) (i.e., ibuprofen, diclofenac, indomethacin, ketoprofen, ketorolac, mefenamic acid, naproxen, piroxicam, celecoxib, and etoricoxib), statins (i.e., atorvastatin, fluvastatin, lovastatin, rosuvastatin, pravastatin, and simvastatin), aspirin, and metformin, and the diagnosis of tobacco use, diabetes, chronic obstructive pulmonary disease (COPD), hypertension, hyperlipidemia, cirrhosis, chronic kidney disease (CKD), peptic ulcer, pneumonia, schizophrenia, depressive disorders, anxiety disorders, bipolar disorders, personality disorders, delusional disorders, other nonorganic psychoses, and dementia[19,30–32]. All confounders were identified by using clinical information between January 01, 2001 and the index date. The coding details for these confounders are shown in Supplementary Table 1.

**Statistical analysis**. We tabulated sample characteristics at the index date for cases and controls. We conducted two multivariable conditional logistic regressions to estimate risks of lung cancer associated with the cumulative use of (1) flupentixol (analysis one); and (2) any antipsychotics (including flupentixol [analysis two]), separately. All confounding variables were adjusted in the models. For the analysis of flupentixol treatment (analysis one), the model was further adjusted for the cumulative use of antipsychotics except for flupentixol (categorized as 0–365 days [selected as the reference group], 366–1825 days, and 1826+ days). All parameters were expressed as odds ratios (ORs) with a 95% confidence interval (95% CI). We carried out subgroup analyses stratified by sex, as there is a sex difference in response to antipsychotic treatment[33,34]. In the secondary analysis, the outcome of lung cancer was further classified into seven subgroups according to the ICD-CM-9 codes, including malignant neoplasm of trachea (ICD-CM-9 code, 162.0), malignant neoplasm of main bronchus (162.2), malignant neoplasm of upper lobe, bronchus or lung (162.3), malignant neoplasm of middle lobe, bronchus or lung (162.4), malignant neoplasm of lower lobe, bronchus or lung (162.5), malignant neoplasm of other parts of bronchus or lung (162.8), and malignant neoplasm of bronchus and lung, unspecified (162.9). We fitted seven models using the leave-one-out approach, which removed one subtype of lung cancer from the outcome each time[35]. We did two sets of sensitivity analyses to test the robustness of study findings: (1) Excluding all patients who had any previous use of risperidone, pimozide, aripiprazole, olanzapine, lurasidone, brexpiprazole, trifluoperazine, clozapine, chlorpromazine, and haloperidol before the index date, as these drugs been suggested to be associated with lung cancer risk reduction;[19,36–44] and (2) For the case-control matching procedure, each patient can be selected as a control for an unlimited number of cases.

We used statistical software R (version 4.1.2) for all analyses[45]. Two-sided *P* values of 0.05 or below were considered indicative of statistical significance.

**Reporting summary**. Further information on research design is available in the Nature Portfolio Reporting Summary linked to this article.

## Results

**Participant characteristics**. We identified 445,015 patients with antipsychotic prescriptions between January 01, 2001, and August 31, 2022. After examining for eligibility, 6435 cases and 64,348

**Table 1 Descriptive statistics of the study sample.**

|                                | Case, no. (%)   | Control, no. (%)  |
|--------------------------------|-----------------|-------------------|
| Patients, no.                  | 6435            | 64,348            |
| Gender                         |                 |                   |
| Female             | 2313 (35.94)    | 23,130 (35.95)    |
| Male               | 4122 (64.06)    | 41,218 (64.05)    |
| Age                            |                 |                   |
| 18–24              | 0               | 0                 |
| 25–44              | 104 (1.62)      | 1040 (1.62)       |
| 45–64              | 1514 (23.53)    | 15140 (23.53)     |
| 65–84              | 3409 (52.98)    | 34090 (52.98)     |
| 85+                | 1408 (21.88)    | 14078 (21.88)     |
| Mean (SD), y       | 73.60 (12.79)   | 73.59 (12.79)     |
| Previous medications           |                 |                   |
| NSAIDs             | 3694 (57.40)    | 31485 (48.93)     |
| Statins            | 1768 (27.47)    | 17729 (27.55)     |
| Aspirin            | 2432 (37.79)    | 25327 (39.36)     |
| Metformin          | 1082 (16.81)    | 12280 (19.08)     |
| Comorbidities                  |                 |                   |
| Tobacco use        | 72 (1.12)       | 502 (0.78)        |
| Diabetes           | 1134 (17.62)    | 12,636 (19.64)    |
| COPD               | 1144 (17.78)    | 6086 (9.46)       |
| Hypertension       | 2205 (34.27)    | 22,995 (35.74)    |
| Hyperlipidemia     | 706 (10.97)     | 6719 (10.44)      |
| Cirrhosis          | 128 (1.99)      | 1113 (1.73)       |
| CKD                | 160 (2.49)      | 1809 (2.81)       |
| Peptic ulcer       | 45 (0.70)       | 417 (0.65)        |
| Pneumonia          | 2188 (34.00)    | 12,507 (19.44)    |
| Schizophrenia      | 645 (10.02)     | 12,795 (19.88)    |
| Depressive disorders | 649 (10.09)   | 8633 (13.42)      |
| Anxiety disorders  | 147 (2.28)      | 2214 (3.44)       |
| Bipolar disorders  | 94 (1.46)       | 2028 (3.15)       |
| Personality disorders | 52 (0.81)    | 783 (1.22)        |
| Delusional disorders | 158 (2.46)    | 3008 (4.82)       |
| Other nonorganic psychoses | 435 (6.76) | 5177 (8.29)     |
| Dementia           | 1106 (17.19)    | 18,218 (28.31)    |

*NSAIDs* non-steroidal anti-inflammatory drugs, *COPD* chronic obstructive pulmonary diseases, *CKD* chronic kidney disease.

matched controls were included for the primary analysis. Of these, 341 patients contributed as both cases and controls and 13,113 patients contributed multiple times in the control groups. Table 1 shows the descriptive statistics for cases and controls. The majority of cases were males ($N = 4122$ [64.06%]), aged 65–84 years (3409 [52.98%]). The mean age of the case group was 74 (SD = 13) years. Compared to controls, cases were more likely to receive a previous prescription for NSAIDs and diagnosis of tobacco use, COPD, cirrhosis, and pneumonia.

The sample characteristics by exposure group of flupentixol and any antipsychotics are shown in Supplementary Table 2. Despite the exposure period, most patients in the flupentixol group were males. Compared to less than 365 days users, patients with longer exposure to flupentixol were younger, more likely to receive NSAIDS and metformin, and had a higher prevalence of cirrhosis, schizophrenia, depressive disorders, bipolar disorders, personality disorders, delusional disorders, and other nonorganic psychoses. The highest prevalence of comorbidity was found in schizophrenia in 366–1825 days ($N = 666$ [73.84%]) and 1826+ days (703 [85.94%]) groups. The descriptive statistics were similar for any antipsychotics.

**Odds of lung cancer associated with antipsychotic use**. Before the index date, 98.85% ($N = 6361$) of cases and 97.34% (62,634) of controls had no or less than 365 days of exposure to flupentixol (Fig. 2). The prevalence of less than 365 days of cumulative use of

any antipsychotics in case and control groups was 74.48% (4793) and 50.30% (32,367), respectively. Controls were more likely to have longer exposure to flupentixol (i.e., 1.42% of controls versus 0.59% of cases for 366–1825 days of exposure; 1.24% of controls versus 0.56% of cases for 1826+ days of exposure) and any antipsychotics (i.e., 27.87% of controls versus 13.13% of cases for 366–1825 days of exposure; 21.83% of controls versus 12.39% of cases for 1826+ days of exposure). Compared to less than 365 days of exposure, patients with 366–1825 days of exposure to flupentixol (OR = 0.65 [95% CI, 0.47–0.91]) and any antipsychotics (0.42 [0.38–0.45]) carried a significantly decreased risk of lung cancer. The risk was also lower for longer-term use of any antipsychotics, with the OR of 0.54 (0.47–0.60) for 1826+ days of exposure (Fig. 2).

For sex-stratified analysis, the association between cumulative use of flupentixol and lung cancer was only significant among the female 366–1825 days users (0.50 [0.27–0.91]). A significantly lower risk of lung cancer was found in 366–1825 days (males: OR = 0.41 [95% CI, 0.36–0.45]; females: 0.43 [0.38–0.49]) and 1826+ days (males: OR = 0.52 [95% CI, 0.46–0.59]; females: 0.57 [0.49–0.67]) of exposure to any antipsychotics for males and females. Estimates from the full analysis are shown in Supplementary Table 3.

**Secondary and sensitivity analyses**. Results from secondary analyses (Table 2) showed a similar risk pattern to the main analysis. For instance, for the total sample, compared to the 0–365 days of exposure to flupentixol, the ORs for 366–1825 days use of flupentixol ranged from 0.60 (95 CI, 0.39–0.91) (excluding malignant neoplasm of upper lobe, bronchus or lung) to 0.69 (0.48–1.00) (excluding malignant neoplasm of lower lobe, bronchus or lung). Regarding any antipsychotics, a significantly decreased risk was observed in patients with both 365–1825 days and 1826+ days of exposure.

Results from sensitivity analyses were consistent with the main findings (Supplementary Tables 4 and 5). For example, for the first sensitivity analysis (Supplementary Table 4), a significantly decreased risk of lung cancer was observed in 366–1825 days and 1826+ days of exposure to antipsychotics. However, the lower risk was only found in patients with 1826+ days of flupentixol exposure compared to their flupentixol-free counterparts.

## Discussion

In this study, we identified a lower risk of lung cancer in patients with more than 1 year of exposure to any antipsychotics. For flupentixol, the significantly reduced risk was only observed in the 366–1825 days exposure group. Similar findings were supported by results from any antipsychotics in both males and females, and flupentixol in females with 0–365 days of exposure. Comparable findings were identified in the secondary analysis of different locations of the tumor.

Previous laboratory results indicate that using A549 cell lines, flupentixol has been shown to reduce the level of Bcl-2 expression, a pro-apoptotic protein located on the mitochondrial membrane which is upregulated by the overactivated AKT protein when PI3K/AKT pathway is hyperactivated[16,46]. AKT inhibits BAD and BAX, the pro-apoptotic Bcl-2 family members inducing cytochrome C leakage and apoptosis. The loss of the Bcl-2 signal reduces the survival of lung cancer A549 cell lines[16]. This current study is among one of the large-scale, real-world studies that investigated the inhibitory effect of flupentixol on lung cancer, which introduced evidence of the unexplored potentials of antipsychotics for cancer prevention. The inverse association of flupentixol or any antipsychotics use with lung cancer observed in this study is consistent with preclinical

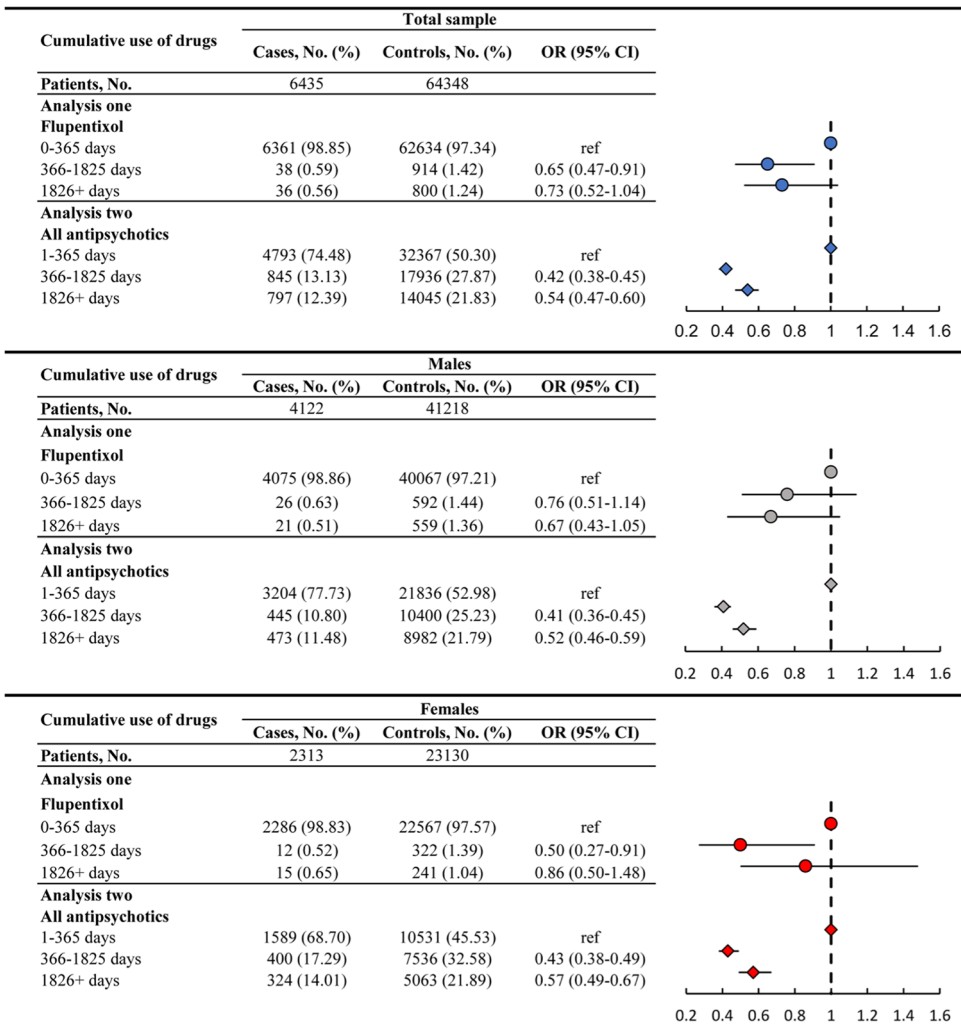

**Fig. 2 Risk of lung cancer associated with cumulative use of flupentixol and any antipsychotics.** Error bars represent the 95% confidence intervals of these odd ratios. Analysis one is to investigate the association between the lung cancer and different cumulative exposure time of flupentixol, adjusting for other antipsychotics and all interested covariates. Analysis two is to investigate the association between the lung cancer and different cumulative exposure time of all antipsychotics (including flupentixol), adjusting for all interested covariates.

evidence from in vitro and in vivo studies showing the potential mechanisms of lung cancer growth suppression on lung cancer[16,47,48]. We showed that in addition to the benefits regarding severe mental illness management on lung cancer, a lung cancer inhibitory effect may also be found in the use of flupentixol and other antipsychotics. Our study added empirical evidence on the choice of different specific antipsychotics to be further explored for potential drug repurposing among people living with severe mental illnesses.

The findings are consistent with a previous real-world case-control study in Mainland China showing an overall protective effect of antipsychotic use on lung cancer compared with non-users[19]. As previous research has suggested a growing trend in antipsychotic prescription worldwide, the risk-benefit analysis of antipsychotic use should indeed be closely updated according to newly generated evidence[32,49]. Similar to flupentixol, reduced risks have been found in other antipsychotics, including trifluoperazine and clozapine, some of which have also been suggested to suppress cancer cells in preclinical studies[42,50]. Indeed, with the increase in accessibility of various antipsychotics, flupentixol, a typical antipsychotic agent, might not be an obvious choice in more recent years, considering the cost and tolerability differences between individuals. Hence, other antipsychotic

agents with promising mechanisms for reducing lung cancer risk should be further explored.

Conversely, there was previous research showing an increased risk of cancer and cancer-related deaths in patients with severe mental illnesses[6,51–53]. However, these studies used the healthy general population as the reference group. In this study, the comparison of lung cancer risk was restricted to patients with a history of antipsychotic medication. The observed reduced risk of lung cancer was a relative risk, determined by comparing antipsychotic users with different exposure times. In addition to antipsychotics, there are many confounders that have a much greater impact on lung cancer incidence, including smoking status, previous radiation therapy, and family history of lung cancer[54]. These lifestyle factors and clinical characteristics can greatly differ between antipsychotic users in this study and the healthy general population in previous research. It is plausible that patients with severe mental illnesses may still have a higher risk of cancer and cancer-related deaths compared to the general population, even considering the potential protective effects of antipsychotics.

There are clear strengths to this study. First, this research investigates the potential of flupentixol in inhibiting lung cancer development within a real-world context, providing

**Table 2 Secondary analyses: risk of lung cancer except for a specific subtype (leave-one-out approach) associated with cumulative use of flupentixol and any antipsychotics.**

| | Total sample | | Males | | Females | |
|---|---|---|---|---|---|---|
| | Analysis one Flupentixol | Analysis two Any antipsychotics | Analysis one Flupentixol | Analysis two Any antipsychotics | Analysis one Flupentixol | Analysis two Any antipsychotics |
| Exclude malignant neoplasm of trachea (162.0) | | | | | | |
| 366–1825 days | 0.65 (0.47–0.91) | 0.42 (0.39–0.45) | 0.76 (0.50–1.13) | 0.41 (0.36–0.45) | 0.50 (0.28–0.91) | 0.43 (0.38–0.49) |
| 1826+ days | 0.74 (0.52–1.04) | 0.54 (0.49–0.60) | 0.67 (0.43–1.05) | 0.52 (0.46–0.60) | 0.87 (0.51–1.49) | 0.58 (0.50–0.67) |
| Exclude malignant neoplasm of main bronchus (162.2) | | | | | | |
| 366–1825 days | 0.67 (0.48–0.94) | 0.42 (0.38–0.45) | 0.78 (0.52–1.17) | 0.41 (0.36–0.45) | 0.51 (0.28–0.92) | 0.43 (0.38–0.49) |
| 1826+ days | 0.75 (0.53–1.06) | 0.54 (0.49–0.60) | 0.70 (0.44–1.09) | 0.52 (0.46–0.60) | 0.86 (0.50–1.48) | 0.57 (0.49–0.67) |
| Exclude malignant neoplasm of upper lobe, bronchus or lung (162.3) | | | | | | |
| 366–1825 days | 0.60 (0.39–0.91) | 0.41 (0.37–0.45) | 0.80 (0.49–1.31) | 0.40 (0.35–0.46) | 0.34 (0.15–0.77) | 0.42 (0.36–0.48) |
| 1826+ days | 0.74 (0.48–1.14) | 0.52 (0.46–0.59) | 0.54 (0.29–1.01) | 0.51 (0.43–0.60) | 1.15 (0.62–2.12) | 0.55 (0.46–0.66) |
| Exclude malignant neoplasm of middle lobe, bronchus or lung (162.4) | | | | | | |
| 366–1825 days | 0.67 (0.48–0.94) | 0.41 (0.38–0.45) | 0.78 (0.52–1.17) | 0.40 (0.36–0.45) | 0.52 (0.29–0.95) | 0.43 (0.38–0.48) |
| 1826+ days | 0.72 (0.51–1.03) | 0.54 (0.49–0.60) | 0.65 (0.41–1.03) | 0.52 (0.45–0.59) | 0.86 (0.50–1.48) | 0.58 (0.50–0.68) |
| Exclude malignant neoplasm of lower lobe, bronchus or lung (162.5) | | | | | | |
| 366–1825 days | 0.69 (0.48–1.00) | 0.43 (0.39–0.46) | 0.75 (0.48–1.17) | 0.40 (0.36–0.45) | 0.61 (0.33–1.15) | 0.46 (0.40–0.52) |
| 1826+ days | 0.67 (0.50–1.01) | 0.53 (0.48–0.59) | 0.69 (0.42–1.13) | 0.52 (0.45–0.60) | 0.64 (0.32–1.29) | 0.55 (0.46–0.65) |
| Exclude malignant neoplasm of other parts of bronchus or lung (162.8) | | | | | | |
| 366–1825 days | 0.65 (0.47–0.91) | 0.42 (0.38–0.45) | 0.76 (0.51–1.14) | 0.41 (0.36–0.45) | 0.50 (0.28–0.90) | 0.43 (0.38–0.49) |
| 1826+ days | 0.74 (0.52–1.04) | 0.54 (0.49–0.60) | 0.68 (0.43–1.06) | 0.52 (0.46–0.59) | 0.86 (0.50–1.48) | 0.57 (0.49–0.66) |
| Exclude malignant neoplasm of bronchus and lung, unspecified (162.9) | | | | | | |
| 366–1825 days | 0.60 (0.37–0.96) | 0.43 (0.38–0.48) | 0.63 (0.45–1.13) | 0.43 (0.37–0.50) | 0.54 (0.25–1.18) | 0.42 (0.35–0.51) |
| 1826+ days | 0.83 (0.54–1.28) | 0.59 (0.52–0.68) | 0.81 (0.48–1.39) | 0.56 (0.47–0.67) | 0.89 (0.44–1.80) | 0.65 (0.52–0.80) |

Analysis one is to investigate the association between the lung cancer and different cumulative exposure time of flupentixol, adjusting for other antipsychotics and all interested covariates.
Analysis two is to investigate the association between the lung cancer and different cumulative exposure time of all antipsychotics (including flupentixol), adjusting for all interested covariates.

supplementary evidence on the properties of antipsychotics in cancer prevention. Second, CDARS serves more than 80% of the entire population in Hong Kong, ensuring the representativeness of our study sample[20]. Third, the diagnoses and prescription records were made by registered doctors using a comprehensive, tertiary-wide electronic health record system. The accuracy and completeness of the diagnostic record in this system have been well validated, and the potential biases introduced by the coding system are thus minimum[55].

This study has some limitations. Firstly, the current dataset only covers patients who used public health care services in Hong Kong. Patients who only presented in the private sector cannot be captured by this study. Secondly, we did not consider different drug dosages in our analyses, which is a possible factor contributing to the magnitude of the association. Thirdly, the under-recording of diagnoses is a common issue in observational studies using electronic health records. For this study, data on diagnosis and medications were not available before 2001. Hence, some estimations, such as the prevalence of comorbidities and the exposure duration of antipsychotics, might be underestimated. Moreover, mental health conditions were associated with surrounded stigma and health seeking behaviors (i.e., bipolar disorder vs. schizophrenia), which might result in the underdiagnosis issue of other health conditions, including the lung cancer. The incomplete capture of comorbidities, especially the mental health problems, may further influence the results. Fourthly, patients with short- and long-term use of flupentixol or antipsychotic showed different demographic and clinical profiles.

Thus, the observed differences in risk profiles for lung cancer and potentially other physical health conditions among patients with varying drug exposure times may be subject to indication bias. Fifthly, the number of patients using flupentixol was modest, which limited the statistical power for a more precise estimate. Sixthly, smoking status and cessation are not routinely recorded in the current dataset[56]. The tobacco use status in this study was identified using ICD-9-CM codes instead of universally self-reported smoking habits, which could lead to an underestimation of the true prevalence of smoking. It is also possible that patients with lung cancer are more likely to be given an ICD-9-CM diagnosis of tobacco use in clinical practice due to more frequent presentations to hospitals. However, given the low observed prevalence of this covariate, any potential bias would be minimal. Furthermore, the lack of smoking cessation data introduces suspicion as to whether the lower risk is attributable to smoking cessation. Nevertheless, a previous study reported an increased smoking rate following the initiation of antipsychotic use, possibly suggesting the use of tobacco as a coping strategy for the side effects of the medication[57]. As such, it is improbable that the observed risk reduction of lung cancer was a result of smoking cessation. Finally, regarding flupentixol, a significantly reduced risk of lung cancer was only observed in certain exposure groups of flupentixol, regardless of the whole sample or the sex-specific groups. Caution is required in the interpretation of the results.

In conclusion, this study showed a decreased risk of lung cancer in patients who used flupentixol or any antipsychotics for more than 1 year. Further research involving other populations

and other specific antipsychotic agents should be conducted to validate the findings and identify more promising antipsychotic agents for potential repurposing in lung cancer prevention.

## Data availability

Data are not available as the data custodians (the Hospital Authority and the Department of Health of Hong Kong SAR) have not given permission for sharing due to patient confidentiality and privacy concerns. Local academic institutions, government departments, or non-governmental organizations may apply for the access to data through the Hospital Authority's data-sharing portal (https://www3.ha.org.hk/data). The numerical data underlying Fig. 2 are shown in tables in Fig. 2. All other data are available from the corresponding author (or other sources, as applicable) on reasonable request.

## Code availability

The code used for this study is available on Zenodo[58] (https://doi.org/10.5281/zenodo.8009776).

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

## Acknowledgements

We gratefully acknowledge the generous provision of data by the Hospital Authority. F.L. and I.W. are partially supported by the Laboratory of Data Discovery for Health (D24H) funded by the by AIR@InnoHK administered by Innovation and Technology Commission.

## Author contributions

Y.C. and F.L. conceptualized and designed the study. I.W. supervised the whole project. Y.H. collected the data. Y.C. made the analysis plan and conducted the statistical analysis. Y.C., R.C., Y.H., I.L., F.C., H.L., M.W., S.C., E.C., I.W., and F.L. interpreted the data. Y.C. and R.C. drafted the manuscript. Y.C., R.C., Y.H., I.L., F.C., H.L., M.W., S.C., E.C., I.W., and F.L. revised the manuscript critically for important intellectual content, gave final approval of the version to be published and agreed to be accountable for all aspects of the work. Y.C., R.C., and Y.H. are co-first authors and I.W. and F. L. share the senior authorship.

## Competing interests

E.C. reports grants from Research Grants Council of Hong Kong, Research Fund Secretariat of the Food and Health Bureau of Hong Kong, National Natural Science Fund of China, Wellcome Trust, Bayer, Bristol-Myers Squibb, Pfizer, Janssen, Amgen, Takeda, and Narcotics Division of the Security Bureau of Hong Kong; honorarium from Hospital Authority; outside the submitted work. I.W. receives research funding outside the submitted work from Amgen, Bristol-Myers Squibb, Pfizer, Janssen, Bayer, GSK, Novartis, the Hong Kong Research Grants Council, the Food and Health Bureau of the Government of the Hong Kong Special Administrative Region, National Institute for Health Research in England, European Commission, and the National Health and Medical Research Council in Australia; has received speaker fees from Janssen and Medice in the previous 3 years; and is an independent non-executive director of Jacobson Medical in Hong Kong. F.L. has been supported by the RGC Postdoctoral Fellowship under the Hong Kong Research Grants Council and has received research grants from the Food and Health Bureau of the Government of the Hong Kong Special Administrative Region, outside the submitted work. The remaining authors declare no competing interests.
