## [Peer Review File · Communications Medicine]

Reviewers' comments:

Reviewer #1 (Remarks to the Author):

In the manuscript of Chai et al., the authors used a nested-case-control design to study the association between psychotics and the risk of lung cancer. The authors reported the risk of lung cancer is halved in relation to use of any antipsychotics 1-4 years before. The authors had careful thoughts on the selection of confounders and did a lot of stratification and sensitivity analyses to support their findings. However, I believe there are a few main flaws in the study design which hinder the validity and interpretation of the results. Therefore I do not find the results convincing. I have a few major comments as below.

1. What is the exposed and what is the unexposed? Is it use of any antipsychotics vs. no use of antipsychotics? Is it use of any flupentixol vs. use of any other antipsychotics? If the first, is the exposed and unexposed groups comparable? Because obviously the exposed group would have the indication (psychosis) while the nonexposed group does not.
2. In methods, Selection of cases and controls, why did the authors restrict "Each patient can contribute as a control for up to four different cases"? By doing so the sampling process is not random and the controls would not be representative of the risk set.
3. In methods, Selection of cases and controls, "Controls were excluded if they did not have any prescription of antipsychotics before the index date ." Why was this performed? By applying this, a lot of unexposed group were excluded from the controls. Then the exposed/unexposed ratio among the controls will be overestimated, which I believe is the reason for having such a dramatically protective effect size.
4. In the second paragraph in the Introduction section, more clarification is needed on the rationale of the specific selection of flupentixol rather than other kinds of antipsychotics as the main interest.

Reviewer #2 (Remarks to the Author):

This is a well written and interesting paper, with appropriate statistical analysis. The methods are well written and appear broadly replicable. The issue of the incidence of cancer and SMI has been well studied, but results continue to be conflicting. The paper therefore adds to this field.

However, while the authors focus on one hypothesis (that antipsychotics may be protective against cancer), I think the manuscript would be strengthened by consideration of alternative hypotheses. I also think the data appears to have some key limitations (very low smoking rate, seeming under-reporting of MH diagnoses) and the impact of these on the findings require further exploration.

More detailed comments below:

Introduction

1. This is a well written and detailed introduction on the potential role of antipsychotics in the suppression of lung cancer. However, I think the paper may benefit from the consideration of

alternative hypotheses (either in the introduction or discussion). For example, some have hypothesised that those with schizophrenia may have a genetic protection to cancer or that because people on antipsychotics die on average earlier than the general population rates of cancer may be lower. Finally, others have suggested that because of stigma, low health literacy and poor access to services in those on antipsychotics, we may be missing cancer in this group, therefore artificially reducing the incidence of cancer in people on antipsychotics.

Methods

2. Index was the first antipsychotic prescription between 2001 and 2022 – does this mean that older patients could have had historical antipsychotic use prior to 2001? I see it is mentioned in the limitations actually, this would benefit from being clearer in the methods.

3. Is there a reason that other psychotic disorders (such as ICD-9: 297-298) are not included? Presumably all on antipsychotics will have a reason for being on them.

4. Do the start and end dates for antipsychotic prescriptions represent a period of continuous supply of antipsychotics? What happens with patients who start and stop antipsychotics multiple times before their cancer diagnosis? Could this be explained in more detail or clarified?

5. Exposure: While this is a detailed section, it would be useful for the reader if the exact exposures were stated. Were these generated a priori, or did you look at the results and then come up with the grouping (less than 1 year, 1-4 etc)? Were the only exposures considered flupentixol and “any antipsychotic”? Listing the exposures would be really helpful.

Results

6. Some information on the constitution of the exposure groups (perhaps as a supplementary table – age, gender, amount of follow up) would also be useful. I am trying to understand why 75% of cases have only had under 1 year of antipsychotic prescription prior to cancer diagnosis, and what these might have been prescribed for. Mental health diagnoses are relatively low in both cases and controls – is this incomplete capture? The fact that only 10% of cases have a diagnosis of schizophrenia surprises me given they all have a prescription of some sort. I am wondering if those with less than 1 years prescription are inherently different to those with longer prescriptions.

7. It might be worth referencing fig 2 at the end of the first paragraph of the results, as it is currently only referenced when talking about raw numbers, but actually this is the only presentation of the main results.

8. Line 196: the 95%CI around the OR of 0.51 are listed as 0.38-0.46, presumably a typo.

9. In eTable2, cases are less likely to have a diagnosis of any of the MH conditions than controls; to a similar degree to longer term antipsychotic use. Could this be because those with less than 1 year antipsychotic use are inherently different, and those with longer term antipsychotic use, and therefore a documented MH condition are more likely to have cancer diagnoses missed? I.e. are the authors sure that what they have here is evidence of the role of antipsychotics, rather than other issues associated with mental health conditions, stigma, access to services etc?

10. In eTable2, why are the results for the main exposure for cumulative use of any antipsychotic not presented, with a – instead? And what does the exposure title “cumulative use of antipsychotics except for flupentixol” pertain to? Is this an error, or does “any antipsychotic” mean any antipsychotic other than flupentixol?

11. Could eTable3 have a clearer title, so readers don’t have to remind themselves what the sensitivity analysis was? And also explained a bit more on line 214?

Discussion

12. It would be good to discuss why you think you found an effect across all antipsychotics. Do you think only some are protective or could there be a shared mechanism related to the drugs?

13. Line 272: I don’t understand this sentence, could you clarify?

14. I feel that the low capture of tobacco use (around 1%, when we would expect high rates in both those with SMI and those with lung cancer) needs more discussion. What is the likely impact of this on your findings? You talk about cessation, and while I agree the impact of that is probably low in the short time period, what about the poor capture of smoking between those with and without lung cancer?

15. The small sample size for the flupentixol is another limitation worth noting.

16. The poor capture of SMI diagnosis, or what you think the indication for these antipsychotics would be worth mentioning.

17. I would like to see more discussion on how your results fit in with the broader evidence of SMI and cancer, re screening rates, missed diagnoses, genetic protection, dying before a point where cancer is common etc.

18. There are some other studies that the authors could consider discussing. For example, an older study found a higher risk of lung cancer but lower risk of other cancers in patients prescribed neuroleptics: <https://www.ncbi.nlm.nih.gov/pmc/articles/PMC2360537/>. While several meta-analyses have found similar/higher rates of cancer in people with schizophrenia, most of whom presumably are on antipsychotics, how do these results fit with this? <https://pubmed.ncbi.nlm.nih.gov/28943096/> and <https://pubmed.ncbi.nlm.nih.gov/30806345/>

19. Many papers find much higher mortality from cancer in people with SMI e.g. <https://pubmed.ncbi.nlm.nih.gov/31660909/>. If antipsychotics are protective, then it is worrying that when diagnosed with cancer, the mortality is high.

Response reviewers' comments

Manuscript Title: “Association of flupentixol and any antipsychotic use with reduced risk of lung cancer: a territory-wide nested case-control study spanning two decades”

We thank the reviewers for the detailed and constructive comments. We have now followed the suggestions and substantially revised the manuscript. Responses to the comments and corresponding revisions are presented below.

Reviewer 1's comments: In the manuscript of Chai et al., the authors used a nested-case-control design to study the association between psychotics and the risk of lung cancer. The authors reported the risk of lung cancer is halved in relation to use of any antipsychotics 1-4 years before. The authors had careful thoughts on the selection of confounders and did a lot of stratification and sensitivity analyses to support their findings. However, I believe there are a few main flaws in the study design which hinder the validity and interpretation of the results. Therefore, I do not find the results convincing. I have a few major comments as below.

Comment 1-1:

What is the exposed and what is the unexposed? Is it use of any antipsychotics vs. no use of antipsychotics? Is it use of any flupentixol vs. use of any other antipsychotics? If the first, is the exposed and unexposed groups comparable? Because obviously the exposed group would have the indication (psychosis) while the nonexposed group does not.

Author Response:

We apologize for the confusion. The initial cohort for our study consists of individuals who received prescriptions for any antipsychotics between 01/01/2001 and 31/08/2022. We included all individuals who had at least one day of exposure to any antipsychotics before the index date, which refers to the date of the first diagnosis of lung cancer for cases and the same date for their matched controls. We actually conducted two separate analyses to investigate 1) the association between incident lung cancer and different cumulative exposure periods of flupentixol, adjusting for other antipsychotics; and 2) the association between incident lung cancer and different cumulative exposure periods of any antipsychotics (including flupentixol). Since all included individuals received antipsychotics but not all of them were flupentixol users, the exposures of these two analyses are thus defined as 1) 0 -365 days (selected as the reference group), 366-1825 days, and 1826+ days of exposure to flupentixol; and 2) 1 -365 days, 366-1825 days,

and 1826+ days of exposure to any antipsychotics (including flupentixol), respectively. We have now added the following clarification in the *Exposure* part in the **Methods** section (Page 8, lines 138-155).

“All patients included in this study had at least one day of exposure to any antipsychotics before the index date. A patient might have multiple records of antipsychotic prescriptions. All antipsychotic prescriptions before the index date were extracted for each patient to calculate the cumulative drug exposure. The exposure period of a single record of the prescription was defined as the time between the start and the end dates of the prescription for that record. The cumulative exposure of all antipsychotics was calculated by summing up the periods of all records of any antipsychotic prescriptions before the index date. If multiple drugs were prescribed within the same period, the exposure period of all antipsychotics would be counted only once. The treatment period of flupentixol was calculated by summing up the periods of all records of flupentixol prescriptions before the index date. According to current clinical guidelines, which suggest the treatment duration of antipsychotics should be one to five years (Takeuchi et al. 2012), the cumulative exposure to any antipsychotics (including flupentixol) was categorized as 1-365 days (selected as the reference group), 366-1825 days, and 1826+ days. Since not all included antipsychotic users received flupentixol, the cumulative exposure of flupentixol was categorized as 0-365 days (selected as the reference group), 366-1825 days, and 1826+ days. Two analyses were conducted separately to investigate the association between the different cumulative exposure times of 1) flupentixol; and 2) any antipsychotics and the reduced risk of incident lung cancer.”

References

Takeuchi, H., Suzuki, T., Uchida, H., Watanabe, K., & Mimura, M. (2012). Antipsychotic treatment for schizophrenia in the maintenance phase: A systematic review of the guidelines and algorithms. *Schizophrenia Research*, *134*(2), 219-225. <https://doi.org/https://doi.org/10.1016/j.schres.2011.11.021>

Comment 1-2:

In methods, Selection of cases and controls, why did the authors restrict "Each patient can contribute as a control for up to four different cases"? By doing so the sampling process is not random and the controls would not be representative of the risk set.

Author Response:

Thank you for your comment. It is true that the density sampling method we employed

for control selection allows a patient to contribute to multiple control subjects without restriction (Vandenbroucke & Pearce, 2012). However, our preliminary results indicate that 918 patients were selected as a control for more than four different cases when each patient was allowed to contribute as a control for an unlimited number of cases. An extreme example is that one patient was selected as a control 19 times. This could potentially decrease the statistical efficiency.² Additionally, there were over 400,000 candidates in the original control pool, which was much larger than the number of cases (N=6,435). The effect of matching with and without replacement on the randomization of the sampling process and the estimated odds ratios is expected to be minimal (Iwagami & Shinozaki, 2022). We ultimately decided to limit the maximum number of times a control participant contributed to the analysis, which is consistent with a previous exemplar study published in *The BMJ* (Hitchings et al. 2022). We have also added a sensitivity analysis in which each patient can be selected as a control for an unlimited number of cases. Results from this sensitivity analysis (Supplement eTable 5) were consistent with the main findings (Supplement page 6).

Reference

- Hitchings, M. D. T., Ranzani, O. T., Lind, M. L., Dorion, M., D'Agostini, T. L., de Paula, R. C., de Paula, O. F. P., de Moura Villela, E. F., Scaramuzzini Torres, M. S., de Oliveira, S. B., Schulz, W., Almiron, M., Said, R., de Oliveira, R. D., Vieira da Silva, P., de Araújo, W. N., Gorinchteyn, J. C., Dean, N. E., Andrews, J. R., Cummings, D. A. T., Ko, A. I., & Croda, J. (2022) Change in covid-19 risk over time following vaccination with CoronaVac: test negative case-control study. *Bmj*, 377, e070102. <https://doi.org/10.1136/bmj-2022-070102>
- Iwagami, M., & Shinozaki, T. (2022). Introduction to Matching in Case-Control and Cohort Studies. *Annals of Clinical Epidemiology*, 4(2), 33-40. <https://doi.org/10.37737/ace.22005>
- Vandenbroucke, J. P., & Pearce, N. (2012). Case-control studies: basic concepts. *Int J Epidemiol*, 41(5), 1480-1489. <https://doi.org/10.1093/ije/dys147>

Comment 1-3:

In methods, Selection of cases and controls, "Controls were excluded if they did not have any prescription of antipsychotics before the index date ." Why was this performed? By applying this, a lot of unexposed group were excluded from the controls. Then the exposed/unexposed ratio among the controls will be overestimated, which I believe is the reason for having such a dramatically protective effect size.

Author Response:

We appreciate this comment. As explained in our response to Comment 1-1, data on patients who were not exposed to any antipsychotics during the study period were not

available to our research team. All patients included in this analysis need to have at least one day of exposure to any antipsychotics before the index date to enable the comparison of lung cancer risks associated with different exposure times of antipsychotic drugs. The index date for controls was the same date as their matched cases. Given that the same patient can be a control for multiple cases, it was possible that the patient did not have any prescription of antipsychotics before the assigned index date and needs to be removed from the pool of controls. We now have added elaborations in the *Selection of cases and controls* part in the **Methods** section (Page 8, lines 134-136).

“For each matching process, control candidates were excluded from the pool of controls if they did not have any prescription of antipsychotics before their assigned index date.”

Comment 1-4:

In the second paragraph in the Introduction section, more clarification is needed on the rationale of the specific selection of flupentixol rather than other kinds of antipsychotics as the main interest.

Author Response:

Thanks for the comment. We concur that the **Introduction** section would benefit from a clearer rationale for selecting flupentixol as the exposure of interest. We have now further highlighted the established safety and effectiveness profile of flupentixol with citations, and thus, the potential for it to be considered an alternative agent if it were found to be particularly associated with a larger lung cancer risk reduction. The revised text is copied below (Page 5, lines 76-81).

“The use of second-generation antipsychotics has been increasingly common in recent years (Matone et al., 2012; Park et al., 2017). However, flupentixol, a first-generation antipsychotic agent, has maintained a stable presence in the clinical treatment of schizophrenia due to its efficacy against negative and affective symptoms and a lower reported rate of severe adverse drug reactions (Grohmann et al., 2014; Ruhrmann et al., 2007). In terms of pharmacology, flupentixol demonstrates promising effects in reducing the risk of lung cancer (Dong et al. 2019).”

References

Dong, C., Chen, Y., Li, H., Yang, Y., Zhang, H., Ke, K., Shi, X. N., Liu, X., Li, L., Ma, J., Kung, H. F., Chen, C., & Lin, M. C. (2019). The antipsychotic agent flupentixol is a new PI3K inhibitor and potential anticancer drug for lung cancer. *Int J Biol Sci*, 15(7), 1523-1532. <https://doi.org/10.7150/ijbs.32625>

- Grohmann, R., Engel, R. R., Möller, H. J., Rütger, E., van der Velden, J. W., & Stübner, S. (2014). Flupentixol use and adverse reactions in comparison with other common first- and second-generation antipsychotics: data from the AMSP study. *European Archives of Psychiatry and Clinical Neuroscience*, 264(2), 131-141. <https://doi.org/10.1007/s00406-013-0419-y>
- Matone, M., Localio, R., Huang, Y. S., dosReis, S., Feudtner, C., & Rubin, D. (2012). The relationship between mental health diagnosis and treatment with second-generation antipsychotics over time: a national study of U.S. Medicaid-enrolled children. *Health Serv Res*, 47(5), 1836-1860. <https://doi.org/10.1111/j.1475-6773.2012.01461.x>
- Park, Y., Huybrechts, K. F., Cohen, J. M., Bateman, B. T., Desai, R. J., Paterno, E., Mogun, H., Cohen, L. S., & Hernandez-Diaz, S. (2017). Antipsychotic Medication Use Among Publicly Insured Pregnant Women in the United States. *Psychiatric Services*, 68(11), 1112-1119. <https://doi.org/10.1176/appi.ps.201600408>
- Ruhrmann, S., Kissling, W., Lesch, O.-M., Schmauss, M., Seemann, U., & Philipp, M. (2007). Efficacy of flupentixol and risperidone in chronic schizophrenia with predominantly negative symptoms. *Progress in Neuro-Psychopharmacology and Biological Psychiatry*, 31(5), 1012-1022. <https://doi.org/https://doi.org/10.1016/j.pnpbp.2007.02.014>

Reviewer 2's comments: This is a well written and interesting paper, with appropriate statistical analysis. The methods are well written and appear broadly replicable. The issue of the incidence of cancer and SMI has been well studied, but results continue to be conflicting. The paper therefore adds to this field.

However, while the authors focus on one hypothesis (that antipsychotics may be protective against cancer), I think the manuscript would be strengthened by consideration of alternative hypotheses. I also think the data appears to have some key limitations (very low smoking rate, seeming under-reporting of MH diagnoses) and the impact of these on the findings require further exploration.

More detailed comments below:

Comment 2-1:

Introduction

This is a well written and detailed introduction on the potential role of antipsychotics in the suppression of lung cancer. However, I think the paper may benefit from the consideration of alternative hypotheses (either in the introduction or discussion). For example, some have hypothesised that those with schizophrenia may have a genetic protection to cancer or that because people on antipsychotics die on average earlier than the general population rates of cancer may be lower. Finally, others have suggested that

because of stigma, low health literacy and poor access to services in those on antipsychotics, we may be missing cancer in this group, therefore artificially reducing the incidence of cancer in people on antipsychotics.

Author Response:

We thank for your suggestions. We agree that placing our study within the context of the long-observed inverse association between schizophrenia, antipsychotics, and cancer incidence would likely strengthen the coherence of our work with the existing literature. We have now added the alternative hypotheses you suggested to the **Introduction** section (Page 5, lines 64-81).

“Previous evidence showed that people living with severe mental illnesses or using antipsychotics might have a lower incidence of some cancers (Correll et al., 2015; Li et al., 2018; Tahir et al., 2014). Apart from a potential direct protective effect from the medications, this may be attributed to several reasons. First, some genetic factors involved in the etiology of schizophrenia may act as protective agents against cancer (Gal et al., 2012). For example, the tumor suppressor gene p53 can reduce the risk of cancer through apoptosis (Gal et al., 2012). Second, long-term antipsychotic use is associated with negative physical consequences, leading to a shorter life expectancy in comparison to the general population (Basciotta et al., 2020). This could contribute to lower cancer incidence. Furthermore, the stigma surrounding mental illnesses and antipsychotic prescriptions, coupled with low health literacy and the deprioritization of mental health care, have created obstacles for individuals seeking medical help (Muhorakeye & Biracyaza, 2021). Consequently, the underdiagnosis of diseases, including cancer, might result in an artificially lowered cancer incidence among those receiving antipsychotic treatments.

The use of second-generation antipsychotics has been increasingly common in recent years (Matone et al., 2012; Park et al., 2017). However, flupentixol, a first-generation antipsychotic agent, has maintained a stable presence in the clinical treatment of schizophrenia due to its efficacy against negative and affective symptoms and a lower reported rate of severe adverse drug reactions (Grohmann et al., 2014; Ruhrmann et al., 2007). In terms of pharmacology, flupentixol demonstrates promising effects in reducing the risk lung cancer (Dong et al. 2019).”

References

Basciotta, M., Zhou, W., Ngo, L., Donnino, M., Marcantonio, E. R., & Herzig, S. J. (2020). Antipsychotics and the Risk of Mortality or Cardiopulmonary Arrest in Hospitalized Adults. *J Am Geriatr Soc*, 68(3), 544-550. <https://doi.org/10.1111/jgs.16246>

- Correll, C. U., Detraux, J., De Lepeleire, J., & De Hert, M. (2015). Effects of antipsychotics, antidepressants and mood stabilizers on risk for physical diseases in people with schizophrenia, depression and bipolar disorder. *World Psychiatry, 14*(2), 119-136. <https://doi.org/https://doi.org/10.1002/wps.20204>
- Dong, C., Chen, Y., Li, H., Yang, Y., Zhang, H., Ke, K., Shi, X. N., Liu, X., Li, L., Ma, J., Kung, H. F., Chen, C., & Lin, M. C. (2019). The antipsychotic agent flupentixol is a new PI3K inhibitor and potential anticancer drug for lung cancer. *Int J Biol Sci, 15*(7), 1523-1532. <https://doi.org/10.7150/ijbs.32625>
- Gal, G., Goral, A., Murad, H., Gross, R., Pugachova, I., Barchana, M., Kohn, R., & Levav, I. (2012). Cancer in parents of persons with schizophrenia: Is there a genetic protection? *Schizophrenia Research, 139*(1), 189-193. <https://doi.org/https://doi.org/10.1016/j.schres.2012.04.018>
- Grohmann, R., Engel, R. R., Möller, H. J., Rütger, E., van der Velden, J. W., & Stübner, S. (2014). Flupentixol use and adverse reactions in comparison with other common first- and second-generation antipsychotics: data from the AMSP study. *European Archives of Psychiatry and Clinical Neuroscience, 264*(2), 131-141. <https://doi.org/10.1007/s00406-013-0419-y>
- Li, H., Li, J., Yu, X., Zheng, H., Sun, X., Lu, Y., Zhang, Y., Li, C., & Bi, X. (2018). The incidence rate of cancer in patients with schizophrenia: A meta-analysis of cohort studies. *Schizophrenia Research, 195*, 519-528. <https://doi.org/https://doi.org/10.1016/j.schres.2017.08.065>
- Matone, M., Localio, R., Huang, Y. S., dosReis, S., Feudtner, C., & Rubin, D. (2012). The relationship between mental health diagnosis and treatment with second-generation antipsychotics over time: a national study of U.S. Medicaid-enrolled children. *Health Serv Res, 47*(5), 1836-1860. <https://doi.org/10.1111/j.1475-6773.2012.01461.x>
- Muhorakeye, O., & Biracyaza, E. (2021). Exploring Barriers to Mental Health Services Utilization at Kabutare District Hospital of Rwanda: Perspectives From Patients. *Front Psychol, 12*, 638377. <https://doi.org/10.3389/fpsyg.2021.638377>
- Park, Y., Huybrechts, K. F., Cohen, J. M., Bateman, B. T., Desai, R. J., Paterno, E., Mogun, H., Cohen, L. S., & Hernandez-Diaz, S. (2017). Antipsychotic Medication Use Among Publicly Insured Pregnant Women in the United States. *Psychiatric Services, 68*(11), 1112-1119. <https://doi.org/10.1176/appi.ps.201600408>
- Ruhrmann, S., Kissling, W., Lesch, O.-M., Schmauss, M., Seemann, U., & Philipp, M. (2007). Efficacy of flupentixol and risperidone in chronic schizophrenia with predominantly negative symptoms. *Progress in Neuro-Psychopharmacology and Biological Psychiatry, 31*(5), 1012-1022. <https://doi.org/https://doi.org/10.1016/j.pnpbp.2007.02.014>
- Tahir, R., Charles, V. C., Virginia, K., John, L., Austin, C., Kari, M., & Robert, S. K. (2014). Antipsychotic Treatment in Breast Cancer Patients. *American Journal of Psychiatry, 171*(6), 616-621. <https://doi.org/10.1176/appi.ajp.2013.13050650>

Comment 2-2:

Methods

Index was the first antipsychotic prescription between 2001 and 2022 – does this mean that older patients could have had historical antipsychotic use prior to 2001? I see it is mentioned in the limitations actually, this would benefit from being clearer in the methods.

Author Response:

Thanks for your suggestion. Yes, patients may have used antipsychotics prior to 2001. However, the clinical information before the year 2001 was not available to our research team. The index prescription of antipsychotics during the study period was thus defined as the first prescription of antipsychotics between 2001 and 2022. We have now added the following clarification in the *Selection of cases and controls* part in the **Methods** section (Page 8, lines 120-122).

“Given the clinical information before 2001 was not available, the first prescription of antipsychotics between 2001 and 2022 was designated as the index prescription of antipsychotics for each patient during the study period. Lung cancer was identified by the ICD-9-CM codes 162.0-162.9...”

Comment 2-3:

Is there a reason that other psychotic disorders (such as ICD-9: 297-298) are not included? Presumably all on antipsychotics will have a reason for being on them.

Author Response:

Thanks for the comments. We assume you refer to the confounders here. We now have added the delusional disorders (ICD-9-CM: 297) and other nonorganic psychoses (ICD-9-CM: 298) as covariates and the results have been updated.

Comment 2-4:

Do the start and end dates for antipsychotic prescriptions represent a period of continuous supply of antipsychotics? What happens with patients who start and stop antipsychotics multiple times before their cancer diagnosis? Could this be explained in more detail or clarified?

Author Response:

We appreciate this comment. Patients continuously received drugs between the start and end dates of each single antipsychotic prescription. A patient can have multiple records of antipsychotic prescription throughout the whole study period. The

cumulative exposure time of antipsychotics was calculated by summing up the durations of all records of antipsychotic prescriptions before the index date. We have now revised the paragraph in the *Exposure* part in the **Methods** section to clarify (Page 8, lines 138-155).

“All patients included in this study had at least one day of exposure to any antipsychotics before the index date. A patient might have multiple records of antipsychotic prescriptions. All antipsychotic prescriptions before the index date were extracted for each patient to calculate the cumulative drug exposure. The exposure period of a single record of the prescription was defined as the time between the start and the end dates of the prescription for that record. The cumulative exposure of all antipsychotics was calculated by summing up the periods of all records of any antipsychotic prescriptions before the index date. If multiple drugs were prescribed within the same period, the exposure period of all antipsychotics would be counted only once. The treatment period of flupentixol was calculated by summing up the periods of all records of flupentixol prescriptions before the index date. According to current clinical guidelines, which suggest the treatment duration of antipsychotics should be one to five years (Takeuchi et al. 2012), the cumulative exposure to any antipsychotics (including flupentixol) was categorized as 1-365 days (selected as the reference group), 366-1825 days, and 1826+ days. Since not all included antipsychotic users received flupentixol, the cumulative exposure of flupentixol was categorized as 0-365 days (selected as the reference group), 366-1825 days, and 1826+ days. Two analyses were conducted separately to investigate the association between the different cumulative exposure times of 1) flupentixol; and 2) any antipsychotics and the reduced risk of incident lung cancer.”

References

Takeuchi, H., Suzuki, T., Uchida, H., Watanabe, K., & Mimura, M. (2012). Antipsychotic treatment for schizophrenia in the maintenance phase: A systematic review of the guidelines and algorithms. *Schizophrenia Research*, *134*(2), 219-225.
<https://doi.org/https://doi.org/10.1016/j.schres.2011.11.021>

Comment 2-5:

Exposure: While this is a detailed section, it would be useful for the reader if the exact exposures were stated. Were these generated a priori, or did you look at the results and then come up with the grouping (less than 1 year, 1-4 etc)? Were the only exposures considered flupentixol and “any antipsychotic”? Listing the exposures would be really helpful.

Author Response:

Thanks for the comment. We actually conducted two separate analyses to investigate 1) the association between incident lung cancer and different cumulative exposure periods of flupentixol, adjusting for other antipsychotics; and 2) the association between incident lung cancer and different cumulative exposure periods of any antipsychotics (including flupentixol). Clinical guidelines suggest that antipsychotic treatment should be continued for 1-5 years (Takeuchi et al. 2012). Since all included individuals received antipsychotics but not all of them were flupentixol users, the exposures of these two analyses are thus defined as 1) 0 -365 days (selected as the reference group), 366-1825 days, and 1826+ days of exposure to flupentixol; and 2) 1 -365 days, 366-1825 days, and 1826+ days of exposure to any antipsychotics (including flupentixol), respectively. We have now added more information on the exposures in the *Exposure* part in the **Methods** section to clarify. Please refer to the revised paragraph which is copied in Comment 2-4.

References

Takeuchi, H., Suzuki, T., Uchida, H., Watanabe, K., & Mimura, M. (2012). Antipsychotic treatment for schizophrenia in the maintenance phase: A systematic review of the guidelines and algorithms. *Schizophrenia Research*, 134(2), 219-225. <https://doi.org/https://doi.org/10.1016/j.schres.2011.11.021>

Comment 2-6:**Results**

Some information on the constitution of the exposure groups (perhaps as a supplementary table – age, gender, amount of follow up) would also be useful. I am trying to understand why 75% of cases have only had under 1 year of antipsychotic prescription prior to cancer diagnosis, and what these might have been prescribed for. Mental health diagnoses are relatively low in both cases and controls – is this incomplete capture? The fact that only 10% of cases have a diagnosis of schizophrenia surprises me given they all have a prescription of some sort. I am wondering if those with less than 1 years prescription are inherently different to those with longer prescriptions.

Author Response:

Thanks for the suggestion and comment. We now have added the eTable 2 in the Supplement to show the sample characteristics by exposure group of flupentixol and any antipsychotics separately. The eTable 2 is copied at the end of this response letter.

We have added the description of eTable 2 in the *Participant characteristics* part in the **Results** section (Page 12, lines 210-217).

“The sample characteristics by exposure group of flupentixol and any antipsychotics are shown in Supplement eTable 2. Despite the exposure period, most patients in the flupentixol group were males. Compared to less than 365 days users, patients with longer exposure to flupentixol were younger, more likely to receive NSAIDs and Metformin, and had a higher prevalence of cirrhosis, schizophrenia, depressive disorders, bipolar disorders, personality disorders, delusional disorders, and other nonorganic psychoses. The highest prevalence of comorbidity was found in schizophrenia in 366-1825 days (N=666 [73.84%]) and 1826+ days (703 [85.94%]) groups. The descriptive statistics were similar for any antipsychotics.”

Similar to other observational studies that use electronic health records without any randomization processes, the current study may be subject to indication bias. The short duration of antipsychotic use could indicate a less severe illness compared to those with long-term use. This hypothesis was supported by the results from Supplement eTable 2, which shows that the prevalence of schizophrenia in patients with 1826+ days of exposure to flupentixol and any antipsychotics is 85.94% and 59.36%, respectively. In addition to schizophrenia, antipsychotics can also be prescribed for other conditions, including depressive disorders, bipolar disorders, anxiety disorders, and dementia. This was supported by eTable 2, in which a relatively higher prevalence of depressive disorders (1-365 days: 11.24%; 366-1825 days: 17.36%) and dementia (1-365 days: 26.12%; 366-1825 days: 43.15%) in patients with a short duration of antipsychotics use was observed. Therefore, the cancer risk profile may differ among patients with varying disease severities or conditions. This could potentially induce a bias towards a higher risk of lung cancer among those with short durations of antipsychotic use, as opposed to the actual results of our study.

We agree that the prevalence of mental illness diagnoses in the total sample is low. There are two possible reasons. First, under-recording of diagnoses is a common limitation in all observational studies using electronic health records. Second, data before the year 2001 are not made available to our research team. Diagnoses of mental illness prior to 2001 cannot be captured, and the prevalence of psychiatric comorbidities might therefore be underestimated. However, it is unlikely a better capture of and adjustment for mental health diagnoses would attenuate the currently observed inverse association between antipsychotics and lung cancer, because no mental health condition is known to be a cause of lower lung cancer incidence.

We have now acknowledged these limitations in the **Discussion** section to caution the readers while interpreting the results (Page 16, lines 305-312).

“Thirdly, the under-recording of diagnoses is a common issue in observational studies using electronic health records. Moreover, data on diagnoses and medications were not available before 2001. Hence, some estimations, such as the prevalence of comorbidities and the exposure duration of antipsychotics, might be underestimated. Fifthly, patients with a short duration of antipsychotic use might indicate a different condition or a less severe illness compared to those with long-term use. This could potentially induce an indication bias towards varied cancer risk profiles among patients with different exposure times of antipsychotics.”

Comment 2-7:

It might be worth referencing fig 2 at the end of the first paragraph of the results, as it is currently only referenced when talking about raw numbers, but actually this is the only presentation of the main results.

Author Response:

Thanks for your suggestion. We now have referenced Figure 2 in the last sentence of the *Odds of lung cancer associated with antipsychotic use* part in the **Results** section (Page 13, line 230).

Comment 2-8:

Line 196: the 95% CI around the OR of 0.51 are listed as 0.38-0.46, presumably a typo.

Author Response:

We apologize for the typo. We have revised the typo and carefully checked all remaining results.

Comment 2-9:

In eTable2, cases are less likely to have a diagnosis of any of the MH conditions than controls; to a similar degree to longer term antipsychotic use. Could this be because those with less than 1 year antipsychotic use are inherently different, and those with longer term antipsychotic use, and therefore a documented MH condition are more likely to have cancer diagnoses missed? I.e. are the authors sure that what they have here is evidence of the role of antipsychotics, rather than other issues associated with mental health conditions, stigma, access to services etc?

Author Response:

Thanks for this comment. The eTable2 has been modified as eTable 3 now. Please refer to our response to Comment 2-6.

Comment 2-10:

In eTable2, why are the results for the main exposure for cumulative use of any antipsychotic not presented, with a – instead? And what does the exposure title “cumulative use of antipsychotics except for flupentixol” pertain to? Is this an error, or does “any antipsychotic” mean any antipsychotic other than flupentixol?

Author Response:

Thanks for this comment. The previous eTable 2 has been modified as eTable 3 now. In fact, eTable 3 shows the results for two separate analyses: 1) the association between the incident lung cancer and the previous cumulative exposure period of flupentixol, adjusting for the cumulative use of other antipsychotics except for flupentixol; and 2) the association between the incident lung cancer and the previous cumulative exposure period of any antipsychotics (including flupentixol). Thus, in the current eTable 3, the estimated effects of cumulative use of other antipsychotics were only available in the analysis of flupentixol (analysis one). We have revised the *Statistical analysis* part in the **Results** section to clarify (Page 10, lines 175-181).

“We conducted two multivariable conditional logistic regressions to estimate risks of lung cancer associated with the cumulative use of 1) flupentixol (analysis one); and 2) any antipsychotics (including flupentixol [analysis two]), separately. All confounding variables were adjusted in the models. For the analysis of flupentixol treatment (analysis one), the model was further adjusted for the cumulative use of other antipsychotics except for flupentixol (categorized as 0-365 days [selected as the reference group], 366-1825 days, and 1826+ days).”

Comment 2-11:

Could eTable3 have a clearer title, so readers don’t have to remind themselves what the sensitivity analysis was? And also explained a bit more on line 214?

Author Response:

Thanks for your suggestion. The previous eTable 3 has been modified as eTable 4. We have revised the title of eTable 4 and also added more information on the results of sensitivity analysis at the end of the **Results** section. The revised parts are copied below

(Page 13, lines 245-249).

“eTable 4. Results from the sensitivity analysis of excluding patients with previous exposure to drugs which were associated with lung cancer risk reduction.”

“Results from the sensitivity analysis were generally consistent with the main findings (Supplement eTables 4 and 5). For example, for the first sensitivity analysis (Supplement eTable 4), a significantly decreased risk of lung cancer was observed in 366-1825 days and 1826+ days of exposure to antipsychotics. However, the lower risk was only found in patients with 1826+ days of flupentixol exposure compared to their flupentixol-free counterparts.”

Comment 2-12:

It would be good to discuss why you think you found an effect across all antipsychotics. Do you think only some are protective or could there be a shared mechanism related to the drugs?

Author Response:

Thanks for your comment. The main objective of this study was to examine the association between flupentixol and lung cancer. Thus, we grouped other antipsychotic agents as a broader category instead of analyzing them individually. Previous evidence has indicated that some other antipsychotic agents (e.g., trifluoperazine and clozapine) may also have a protective effect against lung cancer through similar mechanisms as the flupentixol. We have now included the relevant references to support our findings, as copied below (Page 15, lines 275-277). However, it can be difficult to explore the cause of these similarities or differences in lung cancer risk among various antipsychotic agents without pharmacological experiments. Due to the inherent limitation of observational study and real-world data, future randomized control trial studies are required to investigate the underlying mechanism between lung cancer and different antipsychotic agents.

“Similar to flupentixol, reduced risks have been found in other antipsychotics, including trifluoperazine and clozapine, some of which have also been suggested to suppress cancer cells in preclinical studies (Yeh et al., 2012; Yin et al., 2015). Indeed, with the increase in accessibility of various antipsychotics, flupentixol, a typical antipsychotic agent, might not be an obvious choice in more recent years, considering the cost and tolerability differences between individuals. Hence, other antipsychotic agents with promising mechanisms for reducing lung cancer risk should be further

explored.”

References

- Yeh, C. T., Wu, A. T., Chang, P. M., Chen, K. Y., Yang, C. N., Yang, S. C., Ho, C. C., Chen, C. C., Kuo, Y. L., Lee, P. Y., Liu, Y. W., Yen, C. C., Hsiao, M., Lu, P. J., Lai, J. M., Wang, L. S., Wu, C. H., Chiou, J. F., Yang, P. C., & Huang, C. Y. (2012). Trifluoperazine, an antipsychotic agent, inhibits cancer stem cell growth and overcomes drug resistance of lung cancer. *Am J Respir Crit Care Med*, 186(11), 1180-1188. <https://doi.org/10.1164/rccm.201207-1180OC>
- Yin, Y. C., Lin, C. C., Chen, T. T., Chen, J. Y., Tsai, H. J., Wang, C. Y., & Chen, S. Y. (2015). Clozapine induces autophagic cell death in non-small cell lung cancer cells. *Cell Physiol Biochem*, 35(3), 945-956. <https://doi.org/10.1159/000369751>

Comment 2-13:

Line 272: I don't understand this sentence, could you clarify?

Author Response:

We apologize for the confusion. We have now modified the sentence as follows (Page 17, lines 314-325).

“Lastly, smoking status and cessation are not routinely recorded in the current dataset (Lau et al. 2017). The tobacco use status in this study was identified using ICD-9-CM codes instead of universally self-reported smoking habits, which could lead to an underestimation of the true prevalence of smoking. It is also possible that patients with lung cancer are more likely to be given an ICD-9-CM diagnosis of tobacco use in clinical practice due to more frequent presentations to hospitals. However, given the low observed prevalence of this covariate, any potential bias would be minimal. Furthermore, the lack of smoking cessation data introduces suspicion as to whether the lower risk is attributable to smoking cessation. Nevertheless, a previous study reported an increased smoking rate following the initiation of antipsychotic use, possibly suggesting the use of tobacco as a coping strategy for the side effects of the medication (Matthews et al. 2011). As such, it is improbable that the observed risk reduction of lung cancer was a result of from smoking cessation.”

References

- Lau, W. C. Y., Chan, E. W., Cheung, C. L., Sing, C. W., Man, K. K. C., Lip, G. Y. H., Siu, C. W., Lam, J. K. Y., Lee, A. C. H., & Wong, I. C. K. (2017). Association between dabigatran vs warfarin and risk of osteoporotic fractures among patients with nonvalvular atrial fibrillation. *JAMA*, 317(11), 1151-1158. <https://doi.org/10.1001/jama.2017.1363>
- Matthews, A. M., Wilson, V. B., & Mitchell, S. H. (2011). The role of antipsychotics in smoking and

smoking cessation. *CNS Drugs*, 25(4), 299-315. <https://doi.org/10.2165/11588170-000000000-00000>

Comment 2-14:

I feel that the low capture of tobacco use (around 1%, when we would expect high rates in both those with SMI and those with lung cancer) needs more discussion. What is the likely impact of this on your findings? You talk about cessation, and while I agree the impact of that is probably low in the short time period, what about the poor capture of smoking between those with and without lung cancer?

Author Response:

Thank you for pointing this out. In fact, smoking status and cessation are not routinely recorded in the current data system (Lau et al. 2017). We operationalized tobacco use according to the presence of ICD-9-CM codes instead of universally self-reported smoking habits. It is, therefore, likely that the observed smoking prevalence is lower than the true prevalence. We have now more explicitly acknowledged this issue as a limitation to be cautious about. We have also suggested the possibility that people with lung cancer may be more likely to be given an ICD-9-CM diagnosis of tobacco use. Nevertheless, given the low observed prevalence of this covariate, this potential bias, if any, would be negligible. The added discussion in the *limitation* part in the **Discussion** section is copied below (Page 17, lines 314-325).

“Lastly, smoking status and cessation are not routinely recorded in the current dataset (Lau et al. 2017). The tobacco use status in this study was identified using ICD-9-CM codes instead of universally self-reported smoking habits, which could lead to an underestimation of the true prevalence of smoking. It is also possible that patients with lung cancer are more likely to be given an ICD-9-CM diagnosis of tobacco use in clinical practice due to more frequent presentations to hospitals. However, given the low observed prevalence of this covariate, any potential bias would be minimal. Furthermore, the lack of smoking cessation data introduces suspicion as to whether the lower risk is attributable to smoking cessation. Nevertheless, a previous study reported an increased smoking rate following the initiation of antipsychotic use, possibly suggesting the use of tobacco as a coping strategy for the side effects of the medication (Matthews et al. 2011). As such, it is improbable that the observed risk reduction of lung cancer was a results of smoking cessation.”

References

Lau, W. C. Y., Chan, E. W., Cheung, C. L., Sing, C. W., Man, K. K. C., Lip, G. Y. H., Siu, C. W., Lam, J. K. Y., Lee, A. C. H., & Wong, I. C. K. (2017). Association between dabigatran vs warfarin

and risk of osteoporotic fractures among patients with nonvalvular atrial fibrillation. *JAMA*, 317(11), 1151-1158. <https://doi.org/10.1001/jama.2017.1363>

Matthews, A. M., Wilson, V. B., & Mitchell, S. H. (2011). The role of antipsychotics in smoking and smoking cessation. *CNS Drugs*, 25(4), 299-315. <https://doi.org/10.2165/11588170-000000000-00000>

Comment 2-15:

The small sample size for the flupentixol is another limitation worth noting.

Author Response:

Thank you for the comment. We have now added the effect of the small sample size of the flupentixol users in the *limitation* part in the **Discussion** section (Page 17, lines 312-314).

“Fifthly, the number of patients using flupentixol was modest, which limited the statistical power for a more precise estimate.”

Comment 2-16:

The poor capture of SMI diagnosis, or what you think the indication for these antipsychotics would be worth mentioning.

Author Response:

Thanks for your suggestion. Please refer to our response to Comment 2-6.

Comment 2-17:

I would like to see more discussion on how your results fit in with the broader evidence of SMI and cancer, re screening rates, missed diagnoses, genetic protection, dying before a point where cancer is common etc.

Author Response:

Thanks for your suggestion. We have now inserted an additional brief discussion surrounding these factors in the **Introduction** and **Discussion** sections. Please refer to our responses to Comments 2-1, 2-6, and 2-14.

Comments 2-18 and 2-19:

There are some other studies that the authors could consider discussing. For example, an older study found a higher risk of lung cancer but lower risk of other cancers in patients prescribed neuroleptics:

<https://www.ncbi.nlm.nih.gov/pmc/articles/PMC2360537/>. While several meta-analyses have found similar/higher rates of cancer in people with schizophrenia, most of whom presumably are on antipsychotics, how do these results fit with this? <https://pubmed.ncbi.nlm.nih.gov/28943096/> and <https://pubmed.ncbi.nlm.nih.gov/30806345/>

Many papers find much higher mortality from cancer in people with SMI e.g. <https://pubmed.ncbi.nlm.nih.gov/31660909/>. If antipsychotics are protective, then it is worrying that when diagnosed with cancer, the mortality is high.

Author Response:

Thank you for these two comments and references. We have now cited these studies in a brief additional discussion. We have summarized that previous research has often found the risk of cancer and cancer-related deaths to be higher among patients with severe mental illnesses (SMI). However, these studies frequently compare the SMI population with a healthy general group. In our study, we restricted our comparison of lung cancer to patients with a history of antipsychotic use. The observed reduced risk of lung cancer was a relative risk, determined by comparing antipsychotic users with different exposure times. Although our results showed a protective effect of antipsychotics on lung cancer, other factors that are much stronger predictors of lung cancer (i.e., smoking status, previous radiation therapy, and family history of lung cancer) differ a lot between the antipsychotic users in our study and the healthy general population in these previous references. Therefore, it is not surprising that people with SMI still have a higher risk of cancer and cancer mortality compared to the general population, even with the potential protective effect of antipsychotics. The added discussion is copied below (Page 15, lines 282-293).

“Previous research has found an increased risk of cancer and cancer-related deaths in patients with severe mental illnesses compared to the healthy general population (Dalton et al., 2006; Li et al., 2018; Ni et al., 2019; Zhuo et al., 2019). In this study, the comparison of lung cancer risk was restricted to patients with a history of antipsychotic medication. The observed reduced risk of lung cancer was a relative risk, determined by comparing antipsychotic users with different exposure times. In addition to antipsychotics, there are many confounders that have a much greater impact on lung cancer incidence, including smoking status, previous radiation therapy, and family history of lung cancer (Jyoti et al., 2016). These lifestyle factors and clinical characteristics can greatly differ between antipsychotic users in this study and the healthy general population in previous research. Consequently, it is not surprising that

patients with severe mental illnesses still have a higher risk of cancer and cancer-related deaths compared to the general population, even with the potential protective effect of antipsychotics.”

References

- Dalton, S. O., Johansen, C., Poulsen, A. H., Nørgaard, M., Sørensen, H. T., McLaughlin, J. K., Mortensen, P. B., & Friis, S. (2006). Cancer risk among users of neuroleptic medication: a population-based cohort study. *Br J Cancer*, *95*(7), 934-939. <https://doi.org/10.1038/sj.bjc.6603259>
- Jyoti, M., Matteo, M., Eva, N., Carlo La, V., & Paolo, B. (2016). Risk factors for lung cancer worldwide. *European Respiratory Journal*, *48*(3), 889. <https://doi.org/10.1183/13993003.00359-2016>
- Li, H., Li, J., Yu, X., Zheng, H., Sun, X., Lu, Y., Zhang, Y., Li, C., & Bi, X. (2018). The incidence rate of cancer in patients with schizophrenia: A meta-analysis of cohort studies. *Schizophrenia Research*, *195*, 519-528. <https://doi.org/https://doi.org/10.1016/j.schres.2017.08.065>
- Ni, L., Wu, J., Long, Y., Tao, J., Xu, J., Yuan, X., Yu, N., Wu, R., & Zhang, Y. (2019). Mortality of site-specific cancer in patients with schizophrenia: a systematic review and meta-analysis. *BMC Psychiatry*, *19*(1), 323. <https://doi.org/10.1186/s12888-019-2332-z>
- Zhuo, C., Zhuang, H., Gao, X., & Triplett, P. T. (2019). Lung cancer incidence in patients with schizophrenia: meta-analysis. *Br J Psychiatry*, *215*(6), 704-711. <https://doi.org/10.1192/bjp.2019.23>

eTable 2. Descriptive statistics of the study sample by exposure group for flupentixol and any antipsychotics

	Flupentixol			Any antipsychotics		
	0-365 days	366-1825 days	1826+ days	1-365 days	366-1825 days	1826+ days
Patients. No.	68995	952	836	37160	18781	14842
Gender						
Female	24853 (36.02)	334 (35.08)	256 (30.62)	1210 (32.62)	7936 (42.26)	5387 (36.30)
Male	44142 (63.98)	618 (64.92)	580 (69.38)	25040 (67.38)	10845 (57.74)	9455 (63.70)
Age						
18-24	0	0	0	0	0	0
25-44	1069 (1.55)	60 (6.30)	15 (1.79)	451 (1.21)	390 (2.08)	303 (2.04)
45-64	15707 (22.77)	517 (54.31)	430 (51.44)	6724 (18.09)	4112 (21.89)	5818 (39.20)
65-84	36761 (53.28)	362 (38.03)	376 (44.98)	20772 (55.90)	9446 (50.30)	7281 (49.06)
85+	15458 (22.40)	13 (13.66)	15 (1.79)	9213 (24.79)	4833 (25.73)	1440 (9.70)
Mean (SD). v	73.87 (12.73)	61.87 (10.90)	63.96 (9.28)	75.45 (12.15)	74.32 (13.40)	68.02 (11.97)
Previous medications						
NSAIDs	34256 (49.65)	463 (48.63)	460 (55.02)	18588 (50.02)	8879 (47.28)	7712 (51.96)
Statins	19141 (27.74)	143 (15.02)	213 (25.48)	11280 (30.36)	4587 (24.42)	3630 (24.46)
Aspirin	27489 (39.84)	139 (14.60)	131 (15.67)	16951 (45.62)	7212 (38.40)	3596 (24.23)
Metformin	13020 (18.87)	158 (16.60)	184 (22.01)	7110 (19.13)	3133 (16.68)	3119 (21.01)
Comorbidities						
Tobacco use	557 (0.83)	3 (0.33)	14 (1.71)	316 (0.87)	119 (0.66)	139 (0.96)
Diabetes	13501 (20.10)	123 (13.64)	146 (17.85)	7852 (21.63)	3297 (18.28)	2621 (18.03)
COPD	7139 (10.63)	44 (4.88)	47 (5.75)	4417 (12.17)	1708 (9.47)	1105 (7.60)
Hypertension	24818 (36.95)	159 (17.63)	223 (27.26)	14538 (40.17)	6568 (36.41)	4049 (27.86)
Hyperlipidemia	7301 (10.87)	49 (5.43)	75 (9.17)	4311 (11.87)	1710 (9.48)	1404 (9.66)
Cirrhosis	1205 (1.79)	14 (1.55)	22 (2.69)	728 (2.01)	270 (1.50)	243 (1.67)
CKD	1956 (2.91)	8 (0.89)	5 (0.61)	1506 (4.15)	288 (1.60)	175 (1.20)
Peptic ulcer	457 (0.68)	1 (0.11)	4 (0.49)	269 (0.74)	114 (0.63)	79 (0.54)
Pneumonia	14512 (21.61)	85 (9.42)	98 (11.98)	8366 (23.04)	3904 (21.64)	2425 (16.69)
Schizophrenia	12071 (17.97)	666 (73.84)	703 (85.94)	1355 (3.73)	3458 (19.17)	8627 (59.36)
Depressive disorders	9070 (13.51)	134 (14.86)	78 (9.54)	4079 (11.24)	3132 (17.36)	2071 (14.25)
Anxiety disorders	2321 (3.46)	27 (2.99)	13 (1.59)	1075 (2.96)	757 (4.20)	529 (3.64)
Bipolar disorders	2061 (3.07)	39 (4.32)	22 (2.69)	514 (1.42)	621 (3.44)	987 (6.80)
Personality disorders	799 (1.19)	25 (2.77)	11 (1.34)	255 (0.70)	249 (1.38)	331 (2.28)
Delusional disorders	3038 (4.52)	78 (8.65)	50 (6.11)	866 (2.39)	1186 (6.57)	1114 (7.66)
Other nonorganic psychoses	5383 (8.02)	130 (14.41)	99 (12.10)	2291 (6.31)	1611 (8.93)	1710 (11.77)
Dementia	19245 (28.66)	44 (4.88)	35 (4.28)	9483 (26.12)	7783 (43.15)	2058 (14.16)

NSAIDs: non-steroidal anti-inflammatory drugs; COPD: chronic obstructive pulmonary diseases; CKD: chronic obstructive pulmonary disease

Reviewers' comments:

Reviewer #1 (Remarks to the Author):

Thanks for the efforts to address these comments and I have the answers to my questions. However, I believe the work will benefit from the following revisions.

1. The title was phrased in a way indicating that the exposure is use of medications and the comparison group (unexposed) is never use of medications. Also, the title should not show the direction of the results. Please consider changing it to "Association between cumulative exposure periods of antipsychotics and risk of lung cancer" or similar.

2. Lines 126-128, "Cases were patients aged 18 years or older with newly diagnosed lung cancer between January 01, 2003, and August 31, 2022. " This appears at odds with Figure 1 where the authors first included those who were diagnosed in 2001-2002 as cases and then excluded them. Authors could start the follow-up of outcome since 2003 and do not include those persons from the beginning. Similar comment with excluding controls who did not have any prescription of antipsychotics before the index date - they should not have been included from the beginning. If I understand correctly, the exposure time window is cumulative period of prescription from the first antipsychotic use until the index date, and the outcome is the cancer diagnosis from January 01, 2003 until August 31, 2022. Please simplify and clarify your initial cohort and nested case-control design.

3. Please rephrase "index prescription" to the "first prescription during study period" or similar. Please save the word "index" for outcome only to avoid confusion.

Reviewer #2 (Remarks to the Author):

The revised manuscript deals with the concerns of both reviewers well, and the additional information acts to make the research more replicable.

While this is an interesting piece of research it does have major limitations, which are now described briefly in the discussion.

I have three, related, concerns with the work as it stands.

1. Firstly, the new table e2 shows that those with longer prescriptions are at reduced risk of a range of physical health problems. The fact that this effect is not limited to cancer suggests to me that there may be something specific about the population with longer term antipsychotic use which means they are less likely to have a cancer diagnosis. The very different age profiles of those with less than a years prescription for flupentixol (22% over the age of 85) and those with more than 5 years (1.8% over the age of 85) also suggests that these two groups are very different. It seems to be that those on short term flupentixol may be more likely to be diagnosed with cancer and a whole range of other physical health conditions because they are inherently different to those who are on longer term flupentixol, i.e. the results are due to confounding by indication.

2. While in the introduction the authors state "Previous evidence showed that people living with severe mental illnesses or using antipsychotics might have a lower incidence of some cancers", the

discussion states “it is not surprising that patients with severe mental illnesses still have a higher risk of cancer and cancer-related deaths compared to the general population”. These two points appear to contradict themselves. In the comments to reviewers the authors state that no mental health diagnosis is known to reduce the risk of cancer. While the mental health diagnosis may not be acting directly, people with schizophrenia may have different genetic profiles to those with bipolar disorder, may experience more stigma or have poorer health seeking behaviour. If this is the case then the fact that schizophrenia diagnoses are so unbalanced between the exposure groups (18% in the 0-365 days group compared to 74 and 86% in the other groups) may be influencing the results.

3. Finally, while the discussion reads ““In this study, we identified a lower risk of lung cancer in patients with more than one year of exposure to flupentixol or any antipsychotics”, confidence intervals were wide and included 1 for the flupentixol 1826+ group. While I appreciate this is likely due to lack of power, I do think this is an issue. Furthermore, the authors state “Similar findings were supported by the sex-stratified analysis.”, but again confidence intervals include 1 for flupentixol in men, and the flupentixol 1826+ group for women. While this may have been due to small sample sizes, I am not sure this then supports the statement that the study has “introduced evidence of the unexplored potentials of antipsychotics for cancer prevention.” Or that “a lung cancer inhibitory effect may also be found in the use of flupentixol and other antipsychotics”.

Taking these three points together, I wonder whether the statement that “In this study, we identified a lower risk of lung cancer in patients with more than one year of exposure to flupentixol or any antipsychotics” is fully supported by the study, and would suggest a more cautious interpretation of the results.

Response reviewers' comments

Manuscript Title: “Association between cumulative exposure periods of flupentixol or any antipsychotics and risk of lung cancer: a territory-wide nested case-control study spanning two decades”

We thank the reviewers for the detailed and constructive comments. We have now followed the suggestions and substantially revised the manuscript. Responses to the comments and corresponding revisions are presented below.

Reviewer 1's comments: Thanks for the efforts to address these comments and I have the answers to my questions. However, I believe the work will benefit from the following revisions.

Comment 1-1:

The title was phrased in a way indicating that the exposure is use of medications and the comparison group (unexposed) is never use of medications. Also, the title should not show the direction of the results. Please consider changing it to "Association between cumulative exposure periods of antipsychotics and risk of lung cancer" or similar.

Author Response:

Thank you for your suggestion. We have now changed the title to “Association between cumulative exposure periods of flupentixol or any antipsychotics and risk of lung cancer: a territory-wide nested case-control study spanning two decades”.

Comment 1-2:

Lines 126-128, "Cases were patients aged 18 years or older with newly diagnosed lung cancer between January 01, 2003, and August 31, 2022. " This appears at odds with Figure 1 where the authors first included those who were diagnosed in 2001-2002 as cases and then excluded them. Authors could start the follow-up of outcome since 2003 and do not include those persons from the beginning. Similar comment with excluding controls who did not have any prescription of antipsychotics before the index date - they should not have been included from the beginning. If I understand correctly, the exposure time window is cumulative period of prescription from the first antipsychotic use until the index date, and the outcome is the cancer diagnosis from January 01, 2003 until August 31, 2022. Please simplify and clarify your initial cohort and nested case-control design.

Author Response:

We apologize for the confusion. It is true that the outcome of incident lung cancer diagnosis was identified between January 01, 2003, and August 31, 2022. Given that each patient could contribute as a control for up to four different cases, individuals in the original control pool could have up to four different index dates (i.e., the date of the first diagnosis of lung cancer for their matched cases). Therefore, for each matching process, we excluded control candidates who did not have any antipsychotic prescriptions before their *assigned* index date. Regarding the exposure time window, it is correct that the cumulative exposure time of drug prescription was from the first antipsychotic use until the index date. We have now revised the Figure 1 and the statements in the **Methods** section to clarify. The revised methodology parts are copied below, and the updated Figure 1 is included at the end of this response letter.

Page 7, lines 117 “Patients who 1) had missing values on age, sex, and date of death; 2) had incorrect records (i.e., the death date was before the date of first prescription of antipsychotics); 3) had a diagnosis of lung cancer before or at the date of the first prescription of antipsychotics; or 4) had a diagnosis of lung cancer before January 01, 2003, were excluded. The years 2001 and 2002 were employed as the screening period to exclude non-incident lung cancer cases. Given the clinical information before 2001 was not available, the first prescription of antipsychotics between 2001 and 2022 was designated as the first prescription of antipsychotics during the study period for each patient.”

Page 8, line 125 “The outcome of this study was the first diagnosis of lung cancer between January 01, 2003, and August 31, 2022. Lung cancer was identified by the ICD-9-CM codes 162.0-162.9.”

Page 8, line 133 “Each patient can contribute as a control for up to four different cases, and thus, they could have up to four different assigned index dates.”

Page 9, line 141 “All antipsychotic prescriptions between the first antipsychotic use and the index date were extracted for each patient to calculate the cumulative drug exposure.”

Comment 1-3:

Please rephrase "index prescription" to the "first prescription during study period" or similar. Please save the word "index" for outcome only to avoid confusion.

Author Response:

We appreciate this suggestion. We have now revised all “index prescription” to the “first prescription during the study period” throughout the whole manuscript.

Reviewer 2's comments: The revised manuscript deals with the concerns of both reviewers well, and the additional information acts to make the research more replicable.

While this is an interesting piece of research it does have major limitations, which are now described briefly in the discussion. I have three, related, concerns with the work as it stands.

Comment 2-1: Firstly, the new table e2 shows that those with longer prescriptions are at reduced risk of a range of physical health problems. The fact that this effect is not limited to cancer suggests to me that there may be something specific about the population with longer term antipsychotic use which means they are less likely to have a cancer diagnosis. The very different age profiles of those with less than a years prescription for flupentixol (22% over the age of 85) and those with more than 5 years (1.8% over the age of 85) also suggests that these two groups are very different. It seems to be that those on short term flupentixol may be more likely to be diagnosed with cancer and a whole range of other physical health conditions because they are inherently different to those who are on longer term flupentixol, i.e. the results are due to confounding by indication.

Author Response:

We appreciate your suggestions. We agree that the demographic and clinical characteristics varied significantly among drug exposure groups (as shown in eTable 2), which may indicate that patients with shorter exposure to flupentixol or antipsychotics are inherently different from those with long-term drug use. Thus, the observed differences in lung cancer risk profiles and potentially other physical health conditions among patients with different exposure durations may be subject to indication bias. We have now added the following information in the *limitation* part in the **Discussion** section to clarify (Page 17, line 316).

“Fourthly, patients with short- and long-term use of flupentixol or antipsychotic showed different demographic and clinical profiles. Thus, the observed differences in risk profiles for lung cancer and potentially other physical health conditions among patients with varying drug exposure times may be subject to indication bias.”

Comment 2-2: While in the introduction the authors state “Previous evidence showed that people living with severe mental illnesses or using antipsychotics might have a lower incidence of some cancers”, the discussion states “it is not surprising that patients with severe mental illnesses still have a higher risk of cancer and cancer-related deaths compared to the general population”. These two points appear to contradict themselves. In the comments to reviewers the authors state that no mental health diagnosis is known to reduce the risk of cancer. While the mental health diagnosis may not be acting directly, people with schizophrenia may have

different genetic profiles to those with bipolar disorder, may experience more stigma or have poorer health seeking behaviour. If this is the case then the fact that schizophrenia diagnoses are so unbalanced between the exposure groups (18% in the 0-365 days group compared to 74 and 86% in the other groups) may be influencing the results.

Author Response:

We apologize for the confusion. In the **Introduction**, we aimed to highlight the potential role of antipsychotics in suppressing lung cancer, which is the hypothesis of the current study. However, there has been little agreement to date on the positive or negative association between antipsychotics use and lung cancer. Although findings from this research provided evidence of a decreased risk of lung cancer associated with antipsychotics, some previous studies found an increased risk of cancer and cancer-related deaths in patients with severe mental illnesses. Thus, in the **Discussion** section, we would like to explore the potential reason of this discrepancy.

In our study, the comparison of lung cancer risk was restricted to patients with a history of antipsychotic medication. The reduced risk of lung cancer was observed in patients with longer exposure to antipsychotics compared to short-term users. However, in studies that found an increased risk of cancer and cancer-related deaths, the reference group was the healthy general population. Given the inherent differences between antipsychotic users and the healthy general population, the opposite findings in this study compared to previous research are not surprising. We have now revised the **Introduction** and **Discussion** sections to avoid confusion. The revised parts are copied below.

Page 5, line 61 “Antipsychotic users, who typically live with severe mental illnesses, have a notably higher crude incidence of lung cancer than the general population...**Interestingly, there is analytic** evidence showed that people living with severe mental illnesses or using antipsychotics might have a lower risk of some cancers.”

Page 15, line 285 “**Conversely, there was** previous research **showing** an increased risk of cancer and cancer-related deaths in patients with severe mental illnesses. **However, these studies used the** healthy general population **as the reference group**... **It is plausible** that patients with severe mental illnesses **may** still have a higher risk of cancer and cancer-related deaths compared to the general population, even **considering** the potential protective effects of antipsychotics.”

We apologize for misunderstanding the indirect protective effect of mental health diagnosis on the risk of lung cancer. We have now acknowledged this potential risk pattern of mental illness in the *limitation* part in the **Discussion** section. The revised part is copied below (Page 16, line 308).

“Thirdly, the under-recording of diagnoses is a common issue in observational studies using electronic health records. For this study, data on diagnosis and medications were not available before 2001. Hence, some estimations, such as the prevalence of comorbidities and the exposure duration of antipsychotics, might be underestimated. Moreover, mental health conditions were associated with surrounded stigma and health seeking behaviors (i.e., bipolar disorder vs. schizophrenia), which might result in the underdiagnosis issue of other health conditions, including the lung cancer. The incomplete capture of comorbidities, especially the mental health problems, may further influence the results.”

Comment 2-3: Finally, while the discussion reads ““In this study, we identified a lower risk of lung cancer in patients with more than one year of exposure to flupentixol or any antipsychotics”, confidence intervals were wide and included 1 for the flupentixol 1826+ group. While I appreciate this is likely due to lack of power, I do think this is an issue. Furthermore, the authors state “Similar findings were supported by the sex-stratified analysis.”, but again confidence intervals include 1 for flupentixol in men, and the flupentixol 1826+ group for women. While this may have been due to small sample sizes, I am not sure this then supports the statement that the study has “introduced evidence of the unexplored potentials of antipsychotics for cancer prevention.” Or that “a lung cancer inhibitory effect may also be found in the use of flupentixol and other antipsychotics”.

Taking these three points together, I wonder whether the statement that “In this study, we identified a lower risk of lung cancer in patients with more than one year of exposure to flupentixol or any antipsychotics” is fully supported by the study, and would suggest a more cautious interpretation of the results.

Author Response:

Thank you for the comments. We agree that a significantly reduced risk of lung cancer was only observed in certain exposure groups of flupentixol, regardless of the whole sample or the sex-specific groups. We have now revised the summarized statements in the **Discussion** section and acknowledged this limitation in the *limitation* part. The revised parts are copied below.

Page 14, line 252 “In this study, we identified a lower risk of lung cancer in patients with more than one year of exposure to any antipsychotics. For flupentixol, the significantly reduced risk was only observed in the 366-1825 days exposure group. Similar findings were supported by results from any antipsychotics in both males and females, and flupentixol in females with 0-365 days of exposure.”

Page 17, line 332 “Finally, regarding flupentixol, a significantly reduced risk of lung cancer was only observed in certain exposure groups of flupentixol, regardless of the whole sample or the sex-specific groups. Caution is required in the interpretation of the results.”

Figure 1. Flowchart of the study sample selection

REVIEWERS' COMMENTS:

Reviewer #2 (Remarks to the Author):

Dear authors,

Thank you for the considerations of the reviewer comments. I now feel that all reviewer comments have been addressed sufficiently, and that the manuscript represents an interesting study with well-balanced narrative surrounding the results.